_Article_

# Multi-epitope immunocapture of huntingtin reveals striatum-selective molecular signatures

Joshua L Justice[1,4], Todd M Greco [iD][1,4], Josiah E Hutton[1], Tavis J Reed[1], Megan L Mair[2,3], Juan Botas [iD][2,3] & Ileana M Cristea [iD][1✉]

## Abstract

**Huntington's disease (HD) is a debilitating neurodegenerative disorder affecting an individual's cognitive and motor abilities. HD is caused by a mutation in the huntingtin gene producing a toxic polyglutamine-expanded protein (mHTT) and leading to degeneration in the striatum and cortex. Yet, the molecular signatures that underlie tissue-specific vulnerabilities remain unclear. Here, we investigate this aspect by leveraging multi-epitope protein interaction assays, subcellular fractionation, thermal proteome profiling, and genetic modifier assays. The use of human cell, mouse, and fly models afforded capture of distinct subcellular pools of epitope-enriched and tissue-dependent interactions linked to dysregulated cellular pathways and disease relevance. We established an HTT association with nearly all subunits of the transcriptional regulatory Mediator complex (20/26), with preferential enrichment of MED15 in the tail domain. Using HD and KO models, we find HTT modulates the subcellular localization and assembly of the Mediator. We demonstrated striatal enriched and functional interactions with regulators of calcium homeostasis and chromatin remodeling, whose disease relevance was supported by HD fly genetic modifiers assays. Altogether, we offer insights into tissue- and localization-dependent (m)HTT functions and pathobiology.**

**Keywords** IP-MS; Mediator; Calcium; Thermal Profiling; Mass Spectrometry
**Subject Categories** Neuroscience; Proteomics

## Introduction

Although discovered as the etiological agent of Huntington's disease (HD) in the early 1990's (MacDonald et al, 1993), the precise mechanisms by which polyglutamine (polyQ) expansion within the N-terminal region of HTT causes toxicity in vulnerable brain cells remains an active area of study (MacDonald et al, 1993). Significant progress has been made in understanding HD

pathobiology by defining the molecular landscapes of HD models using quantitative omics approaches. For example, transcriptomic disturbances are key features of HD-induced pathobiology (Kumar et al, 2014), with recent efforts aimed at teasing apart the molecular contributions of polyQ versus uninterrupted CAG repeat lengths in striatum-enriched pathogenesis (Gu et al, 2022). Moreover, the granularity afforded by single cell/nuclear RNAseq is starting to reveal the contribution of different cell types and neuronal pathways to disease pathobiology (Matsushima et al, 2023; Pressl et al, 2024; Mätlik et al, 2024). Multiomics interrogation of an allelic series of knock-in HD mice (Zheng et al, 2012) has identified sequential polyQ-dependent disruptions of RNA, microRNA, and protein networks (Langfelder et al, 2016, 2018). Further integrative analysis of the allelic series datasets revealed consistent striatal-specific HD-associated disease signatures that are linked to disease progression (Obenauer et al, 2023). Evaluating these molecular signatures in conjunction with in vivo neuronal survival data helped to define cell type-specific temporal dynamics of homeostatic versus pathogenic responses in the striatum (Megret et al, 2021). HD models with perturbed omics networks have been complemented by the development and application of HD modifier screens in mice and flies, identifying genetic loci that affect disease progression (Wertz et al, 2020; Romero et al, 2008; Al-Ramahi et al, 2018; Miller et al, 2012). Therefore, an ongoing area of interest is focused on identifying striatal molecular signatures that are primary versus compensatory drivers of increased vulnerability to HD-induced pathology.

One hypothesis is that high-value disease-relevant candidates are proteins that interact with HTT in the striatum and are linked to HD pathogenesis and/or disease progression (Goehler et al, 2004; Kaltenbach et al, 2007; Neri, 2011; Shirasaki et al, 2012; Greco et al, 2022). While most HTT protein interactions that are disease modifiers have been defined in HD model organisms, there are several interactions identified as human genetic HD modifiers from genome-wide association studies, including TCERG1 (Holbert et al, 2001; Kaltenbach et al, 2007; Andresen et al, 2007; Lobanov et al, 2022), MSH3 (Shirasaki et al, 2012; Flower et al, 2019), and more recently MLH1-PMS2 (Lee et al, 2017; Sun et al, 2024) and POLD1 (Ripaud et al, 2014; Consortium et al, 2024). Over the last 20 years, protein interaction studies of HTT have evolved in concert with core technological advances and the development of increasingly

[1]Department of Molecular Biology, Princeton University, Washington Road, Princeton, NJ 08544, USA. [2]Jan and Dan Duncan Neurological Research Institute, Houston, TX, USA. [3]Department of Molecular and Human Genetics, Baylor College of Medicine, Houston, TX, USA. [4]These authors contributed equally: Joshua L Justice, Todd M Greco. ✉E-mail: icristea@princeton.edu

powerful HD model systems (Silva Ramos et al, 2024). Computational platforms have been developed and made accessible to the community as tools for data integration and visualization, such as the HDinHD portal for HD multi-omic data and model systems and the HTT-OMNI for protein interaction analysis and visualization (Aaronson et al, 2021; Kennedy et al, 2022; Meem et al, 2023). Together, curation of HTT interacting proteins (HIPs) has produced a compilation of >3000 unique candidates (Aaronson et al, 2021). Ironically, the breadth of these interactions has created challenges in identifying HIPs that are disease-relevant and promising candidates for therapeutic targeting. Thus, progress towards identifying proximal, disease-relevant HIPs will require tighter integration of multifaceted experimental and computational approaches. These approaches would leverage the latest HD animal model systems and human specimens to prioritize HIPs through the integration of disease-dependent parameters, such as polyQ dependence, interaction stability and interface, tissue/cell type and subcellular localization, bulk and single-cell multiomics measurements, and modifiers of disease phenotypes. Towards these goals, our group has previously used quantitative proteomics to characterize the impact of expanded polyQ on the formation and relative stability of HIPs in the striatum (Greco et al, 2022) and cortex (Kennedy et al, 2022) in an age-depended manner in mice expressing FLAG-tagged HTT with expanded (Q140) or normal (Q20) polyQ. We found extensive polyQ-dependent HIP dysregulation occurring at an early age, prior to the onset of Q140-dependent HD phenotypes, i.e., as early as 8 weeks old in the striatum. Using bioluminescence-based two-hybrid assays, we showed that the dysregulated HIPs that we found in mouse HD models can also have direct interactions with HTT in human cells. To assess these dysregulated HIPs in the context of disease pathogenesis, a fly HD model showed, at the genetic level, a link between these interacting proteins and altered motor performance. Furthermore, comparing the molecular signatures of HD disease progression observed in the allelic series transcriptome and proteome networks, our study suggested that polyQ-dependent modulation of the interactome precedes the disruption of transcriptome and proteome (Greco et al, 2022).

Encouraged by the findings described above supporting the idea that HIPs are key molecular players in mediating mutant HTT (mHTT) pathobiology, attention should be turned to several fundamental aspects of the HIP landscape that remain unclear. First, we lack a complete understanding of the subset of HIPs that preferentially occur in regions of the brain most impacted by disease (striatum and cortex) compared to regions that are relatively spared of pathology (e.g., the cerebellum). An early pioneering study by Shirasaki and colleagues used immunopurification-based proteomics to identify mHTT-dependent interactions and reveal functional modules associated with the striatum, cortex, and cerebellum interactions (Shirasaki et al, 2012). Yet, the available proteomic technologies did not afford direct quantitative comparisons of HIP abundances between tissues. Second, protein interactions identified with endogenous, non-tagged HTT are vastly underrepresented in the HIP knowledgebase (Kennedy et al, 2022; Aaronson et al, 2021), providing an unknown level of bias in current candidate HIPs. Third, the interaction interfaces of proteins with full-length HTT have not been well characterized. Our recent efforts to define direct HTT interactions using two-hybrid dual bioluminescence assays

highlighted the potential for spatial segregation of HIPs between the N- versus C-terminal domains of HTT. The development of distinct HD pathologies between different brain regions and the availability of HTT antibodies that cover unique epitopes provides an unexplored path to localize HIPs to specific HTT domains within different tissues.

Towards addressing these gaps in knowledge, we integrated multi-epitope protein interaction studies, subcellular fractionation, thermal proteome profiling, and genetic modifier assays to characterize tissue-specific and epitope-enriched protein interactions of endogenous wild-type and mutant HTT. Leveraging human cell, mouse, and fly models allowed us to profile distinct subcellular pools of HTT complexes, which were linked to dysregulated cellular pathways involved in calcium homeostasis, chromatin remodeling, and transcription. Tissue-enriched HIPs were annotated to non-overlapping cellular processes. We also found an HTT association with nearly all subunits of the canonical Mediator complex (20/26), a major transcriptional regulator, with preferential enrichment of the MED15 subunit in the tail domain. The subcellular localization of this interaction was differentially modulated by mHTT in the striatum of HD mice at an early versus later stage of HD-like pathology, while loss of HTT in human neuroblastoma cells altered the subunit assembly state of Mediator. Lastly, we demonstrated that the striatal-enriched HTT interaction with voltage-dependent calcium channels (CACNB and CACNA1) and chromatin remodeling factors, SMARCA5 and CHD2, were modifiers of mHTT phenotypes in HD fly genetic assays. Taken together, this study highlights tissue- and localization-dependent (m)HTT functions that identify candidate sensitizing or protective molecular factors during disease progression.

## Results

### Mapping brain region-dependent HIPs using multi-epitope immunocapture

We sought to characterize the protein interactions of endogenous HTT by considering several layers of regulation in the context of HD—tissue type, polyQ- and age-variation, as well as epitope selectivity (Fig. 1A). Striatum, cortex, and cerebellum were dissected from 8-week and 40-week old HD mice containing a humanized HTT exon 1 with either a normal (Q20) or expanded (Q140) polyQ region. This expanded polyQ length causes progressive behavioral and neuromotor dysfunction in mice, similar to that observed in human disease (Menalled et al, 2003; Hickey et al, 2008). To assess epitope selectivity, endogenous HTT was isolated with antibodies that target different regions of the protein. Individually and in combination, we tested nine anti-HTT antibodies that target the N-terminus, proline-rich region, the helical HEAT repeats, the bridge domain (located between the third and fourth HEAT domains), or the C-terminus (Fig. EV1A). We observed several high-performing antibodies for HTT capture, such as EPR5526, whose epitope lies in the first 17 amino acids (N17) (Fig. EV1A). In contrast, the C-terminal antibodies, 138 and 139, showed lower capture of HTT (Fig. EV1A). For several antibodies with neighboring epitopes, their IP capture performance was additive when used in combination (Fig. EV1B). Specifically, we optimized IP conditions for antibody combinations that targeted

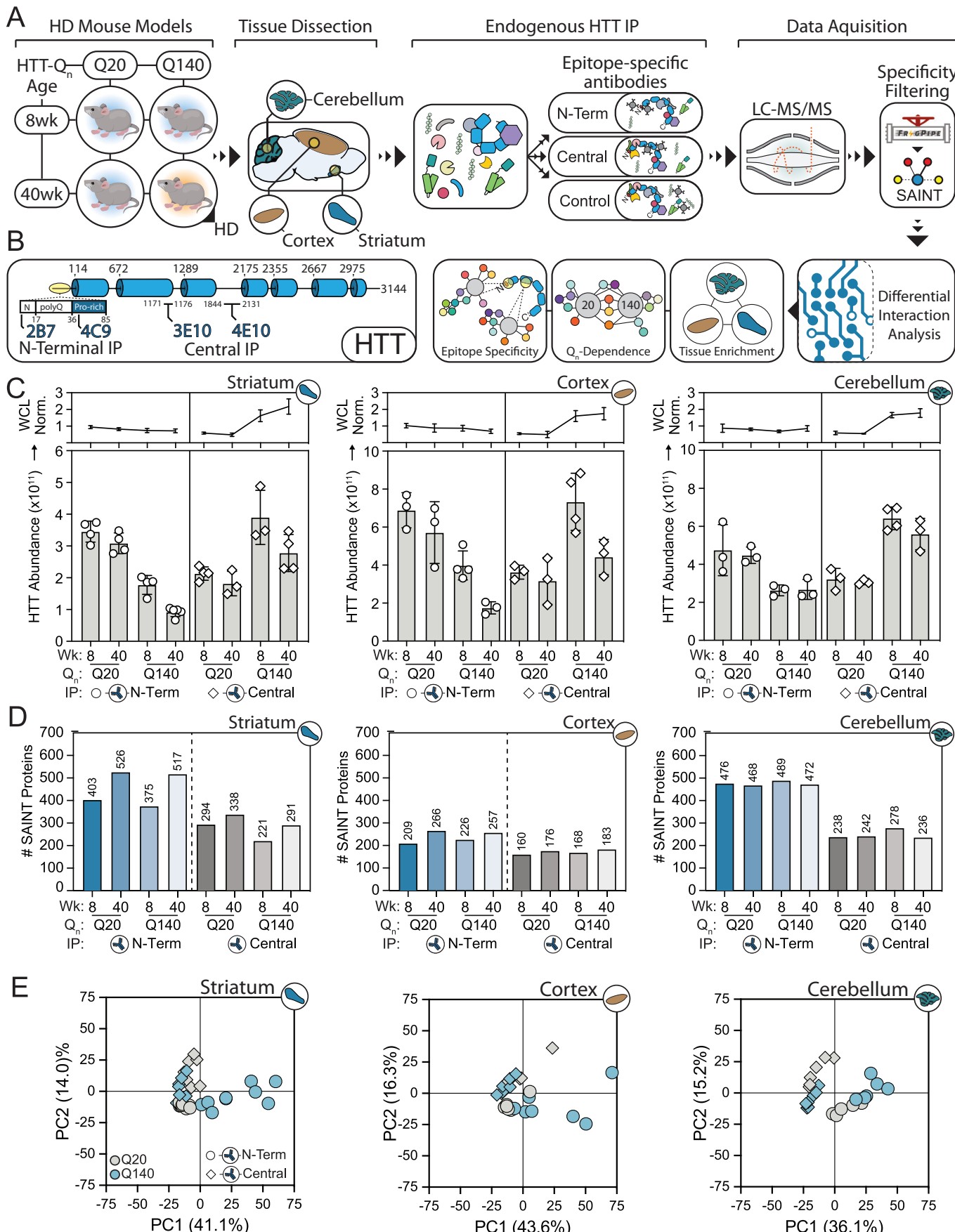

**Figure 1. Characterizing protein interactions of endogenous WT and mutant HTT in HD mice.**

(A) IP-MS strategy and computational analysis workflow. Mice aged 8 weeks or 40 weeks from two genotypes (Q20 and Q140 HTT) were chosen as a model for HD. The striatum, cortex, and cerebellum were dissected from sacrificed animals and lysed. Antibodies that target different regions of HTT (2B7 & 4C9: N-terminus; 3E10 & 4E10: Central) and isotype-matched control antibodies (IgG) were utilized for IP followed by LC-MS/MS. Interaction specificity was determined by SAINT analysis. Precursor-based label-free quantification and bioinformatic analyses were performed to characterize age, genotype, tissue, and epitope-driven HTT interactions. (B) Domain map of HTT with antibody epitope regions denoted. (C) IP-MS protein abundance levels of HTT, isolated from the striatum, cortex, and cerebellum (mean ± SD, $n = 3–5$ biological replicates). HTT abundance was also scaled to the whole-cell lysate amounts of HTT quantified by DIA-MS (line plots, *top*). (D) The number of HTT interactions in each mouse genotype and age that passed SAINT specificity filtering for striatum, cortex, and cerebellum IP conditions. (E) PCA analysis of HTT interacting protein abundances quantified by IP-MS in striatum, cortex, and cerebellum. Interaction abundances were normalized to respective HTT bait abundances.

the N-terminus and proline-rich regions (2B7 and 4C9 antibodies, respectively), as well as the second HEAT repeat and bridge regions (3E10 and 4E10 antibodies, respectively) using western blotting analysis of Q20 whole brain lysates. Next, we evaluated antibody specificity using immunoaffinity purification paired with tandem mass spectrometry (IP-MS) (Dataset EV1). Consistent with the western blot analysis, C-terminal antibodies 138–139 showed ~50–100× lower capture of HTT. The captured proteins with the highest abundance were aquaporin-4 (Aqp4) and Hsp10 (Hspe1), which were uniquely identified in the 138–139 IPs (Fig. EV1C; Dataset EV1), suggesting potential off-target capture. For the isolations with EPR5526, 2B7-4C9, or 4E10-3E10 antibodies, HTT and its known interacting partner, HAP40 (F8a1) were consistently isolated and among the most abundant captured proteins (Fig. EV1C). However, we observed that EPR5526 captured a unique population of interacting proteins compared to 2B7-4C9 and 4E10-3E10 (Fig. EV1C,D). We investigated this further in the striatum and found that, despite the efficient capture of HTT/HAP40 in an ~1:1 ratio, EPR5526 also captured synaptojanin-1 (Synj1) with ~10× higher abundance than HTT/HAP40 (Fig. EV1E). Interestingly, the capture may be based on a shared conformational epitope. The primary sequence alignment of HTT-Synj1 and the Synj1 crystal structure (AlphaFold prediction) supported an alpha-helix conformation (Fig. EV1F), but we did not observe an anti-EPR5526 immunoreactive band in the expected region (~173 kDa) by western blotting. Overall, based on our IP efficiency and specificity analyses, we proceeded with 2B7-4C9 and 4E10-3E10 antibodies for the complete workflow (Fig. 1A). Both of these antibody combinations showed effective capture of HTT relative to the isotype-matched controls for Q20 and Q140 genotypes from the whole brain (Fig. EV1C) and striatum (Fig. EV1D,G), as well as the enrichment for validated HTT interacting proteins (Fig. EV1G), including HAP40.

Following antibody optimizations and specificity assessment, we proceeded to characterize HTT interactions from HD and control mice striatal, cortical, and cerebellar tissues collected at 8 and 40 weeks of age. Tissue lysates were divided between IPs performed with optimal N-Terminal (N-Term) and Central HTT antibodies (Fig. 1B) and isotype-matched IgG controls. In parallel, a portion of the lysate was reserved for label-free proteomics to control for proteome abundance changes of HTT and its interactions, which could influence the interpretation of interaction levels. For example, polyQ expansion resulted in diminished Triton-soluble HTT at the whole proteome level, which was exacerbated by mouse age. This phenotype was stronger in the striatum and cortex (~60% decrease relative to Q20) relative to the cerebellum tissue (~30% decrease). Subsequently, immunoisolates were digested and

analyzed by tandem mass spectrometry as above. We first assessed HTT capture across all conditions. In the N-term IPs, we saw a polyQ- and age-dependent reduction of captured HTT. While most of this effect likely resulted from the lower HTT at the proteome level, it is also possible that the expanded polyQ contributes to the reduced capture performance of the N-term targeting antibodies (Fig. 1C, left panels). Also noteworthy, targeting Central epitopes upon polyQ expansion (Q140) resulted in greater capture of HTT relative to Q20, which cannot be explained by changes in HTT protein abundance. Rather, this may be contributed by changes in epitope availability due to conformational or PPI-driven effects of polyQ expansion. Next, as a positive control, we evaluated the co-isolation of HAP40, which directly interacts with HTT and is the only structurally characterized PPI to date (Guo et al, 2018; Harding et al, 2021). HTT-associated HAP40 levels were reproducibly quantified across all conditions (age, genotype, tissue, and antibody). HAP40 abundance correlated linearly with HTT abundance and was independent of antibody, tissue, or genotype (Fig. EV1H).

When considering the abundance profiles of HTT co-isolated proteins, these were primarily separated by tissue (Fig. EV1I) and antibody (Fig. EV1J), as shown by Principal Component analyses. Further filtering by interaction specificity using the Significant Analysis of INTeractome (SAINTexpress) (Teo et al, 2014) tool identified ~1100 high-confidence HTT interacting proteins as a union across all conditions. This computational tool models the MS signal distribution of proteins from the control (IgG) IPs and the experimental (HTT) IPs to define experiment-optimized thresholds for removal of non-specific interactions. The overall number of SAINT-filtered proteins showed a polyQ-dependent effect that correlated with the known relative tissue vulnerabilities (Fig. 1D). Specifically, a polyQ-dependent increase in the number of SAINT-specific interactions was observed in the striatum, which was much less pronounced in the cortex, while the number of interactions was relatively constant in the cerebellum (Fig. 1D). Moreover when considering the abundance profiles of SAINT-filtered HTT interacting partners isolated from the striatum and cortex, the Q140 40-week-old mice were more distinct when using HTT antibodies targeting the N-terminal versus the Central epitope, as shown by this sample group's variance in PC1 (Fig. 1E, blue vs. gray). In contrast, HTT interaction profiles in the cerebellum showed the most separation by the antibody variable, but less so by the mHTT or age variables (Fig. 1E, diamonds vs circles). Overall, these results suggest that different subsets of proteins interacting with wild-type or mutant HTT may cause preferential immuno-capture depending on the targeted epitopes. Taken together with our observation that mHTT appears to alter antibody affinity

differentially between N-term and Central antibody capture (Fig. 1C), our approach targeting multiple HTT epitopes can provide a more complete perspective of tissue-specific interactions and dysregulated PPIs.

## Epitope-driven HTT PPI signatures reveal tissue and polyQ-dependent cellular functions

Given our finding of epitope-driven PPI signatures, we asked which PPIs were the primary drivers of these unique profiles, whether specific biological processes are enriched in these signatures, and if they were influenced by tissue and polyQ expansion. PPI abundance can vary as a factor of the amount of HTT captured and contributions from biological influences at the proteome level (Fig. 1C). Therefore, we scaled PPI abundances within each tissue based on the HTT level in each IP. Next, we calculated fold changes and the significance of PPI relative abundance changes between the N-terminal and Central region IPs within each mouse genotype and age for each tissue. This comparison allowed us to identify a core set of ~700 consistent HIPs, as well as distinct sets of HIPs that exhibit a preference for specific epitopes. A range of 209–295 HIPs (between tissues) demonstrated preference ($>\log_2$ 2.5) for the N-terminal IP condition, while between 108–130 PPIs preferentially enriched with the Central region IP (Fig. 2A–C). As a next step, we characterized enriched features of the epitope-influenced HIPs across tissues. We first performed gene ontology (GO) enrichment for subcellular localization of the HTT interacting proteins. In the striatum, the N-terminal antibodies enriched for proteins associated with the cytoplasm, synapse, and cytoskeleton (Fig. 2D), which are among the well-represented annotations of previously reported HIPs (Kennedy et al, 2022). Surprisingly, the PPIs enriched from the cortex and cerebellum by the N-terminal IP showed a different localization annotation pattern. The enrichment in localization to the synapse, cell projection, and cell junction were all diminished in the cortex (Fig. 2E) and cerebellum (Fig. 2F)—with the largest decrease in these terms associated with the cerebellum. Instead, in these tissues, the N-terminal IPs enriched for HIPs associated with the nucleus, which was more prominent in the cortex than the cerebellum (Fig. 2E,F, *N-Term*). The nuclear annotation was missing from the N-terminal IP in the striatum but curiously was recovered in the Central IPs (Fig. 2D). These results suggest that the complement of HIPs may be highly tissue-specific and would require access to different epitopes for their capture by antibody-based approaches.

We next asked if the epitope-driven HTT PPI signatures were associated with distinct molecular functions. Using GO analysis to analyze PPIs captured by the N-terminal IPs, we observed a strong enrichment for terms associated with cell division, cell cycle regulation, metabolism, and vesicle trafficking, which were largely shared across all tissues (Fig. 2G). The Central IP condition correlated with protein biosynthesis, metabolism, and DNA repair for all tissue types (Fig. 2H). Interestingly, the cortex and cerebellum displayed an N-terminal IP enrichment for RNA metabolism, transcription, and processing that was missing in the striatum for this IP condition (Fig. 2G) but was present in the striatum for the Central IP condition (Fig. 2H). Our observations of tissue-dependent PPI epitope preferences suggest that a unique assembly of interactions associated with transcription and RNA processing are present in the striatum relative to the other studied

tissues. This is supported by genetic findings that transcriptional regulation is differentially impacted in the striatum in HD (Gu et al, 2022; Kumar et al, 2014).

Upon examination of epitope-specific HTT interactions, we observed a cohort of PPIs that were represented in all the studied tissues. We assembled these PPIs into a STRING-based interaction network, which was explored for functional enrichment and polyQ dependence (Fig. 2I). The N-terminal IPs consistently enriched for a set of proteins associated with acetyl-CoA biosynthesis, including DLAT, PDHA1, PDHB, and PDHX, with an increase in interaction between 8 and 40 weeks in the Q140 HD model genotype. The increase in interaction for these proteins was observed in striatum and cortex tissue, but not in the cerebellum. Conversely, potassium channel proteins showed a strong incidence with the N-terminal IP condition and an increased abundance in the Q140 condition, with the trend present in all studied tissues. The Central IPs revealed associations with a subset of proteins annotated to unfolded protein responses, including HSPA4 and five members of the chaperonin complex. As we observed higher levels of Q140 HTT being isolated by the Central IPs (Fig. 1C), the increased presence of chaperonins may reflect the isolation of a misfolded or aggregated form of HTT. However, the chaperonin association was not increased with Q140 versus the Q20 genotype. Previous studies have found interactions between HTT and heat shock proteins, such as HSPA8 and HSP90 (Shirasaki et al, 2012; Culver et al, 2012; Kim et al, 2016), both of which were represented in our IP-MS dataset. We also observed a high level of interaction between HTT and histone methylation factors, including the entire MLL3/4 complex (KMT2C, KDM6A, PAGR1, PAXIP1, and NCOA6), which was lost with mHTT in the cerebellum. Another striking epitope-selective HIP was CREB-binding protein (CREBBP). In the striatum, its interaction was confirmed in the Central but not N-Terminal IP, yet its detection in the cortex and cerebellum was epitope-independent, which we confirmed by targeted mass spectrometry (Fig. 2J). To support that the divergence of HIPs (Fig. 2A–C) and their annotated localizations (Fig. 2D–F) and functions (Fig. 2G,H) was not driven by an off-target effect, we visualized the subcellular localization of HTT using confocal immunofluorescence. In human neuroblast BE(2)-C cells, the 2B7, 4C9, 3E10, and 4E10 antibodies showed cytoplasmic and nuclear staining, but with clear preferences in the staining distributions (Fig. 2K). For example, the HTT localization detected by the 3E10 antibody was predominantly nuclear or perinuclear, consistent with the Central IP being enriched in transcriptional processes in the striatum. In contrast, the 4C9 (and to lesser extent 2B7) showed largely cytoplasmic staining, with enhanced staining at the cell membrane (Fig. 2K), consistent with the functional enrichment of interactions from the N-terminal HTT IPs in the striatum (Fig. 2D). Taken together, our epitope IP analyses allowed us to identify distinct populations of HTT complexes that may be dictated by tissue-specific functional or structural determinants.

## Endogenous HTT isolation enriches for whole protein complexes modulated by polyQ and tissue type

From our analysis of epitope-enriched HIPs, we noticed many components of multi-subunit protein complexes. For example, we isolated several members of the SEPTIN ring complex with the N-terminal epitope condition and the entire MLL3/4 complex with the Central epitope condition. Given this observation, we analyzed

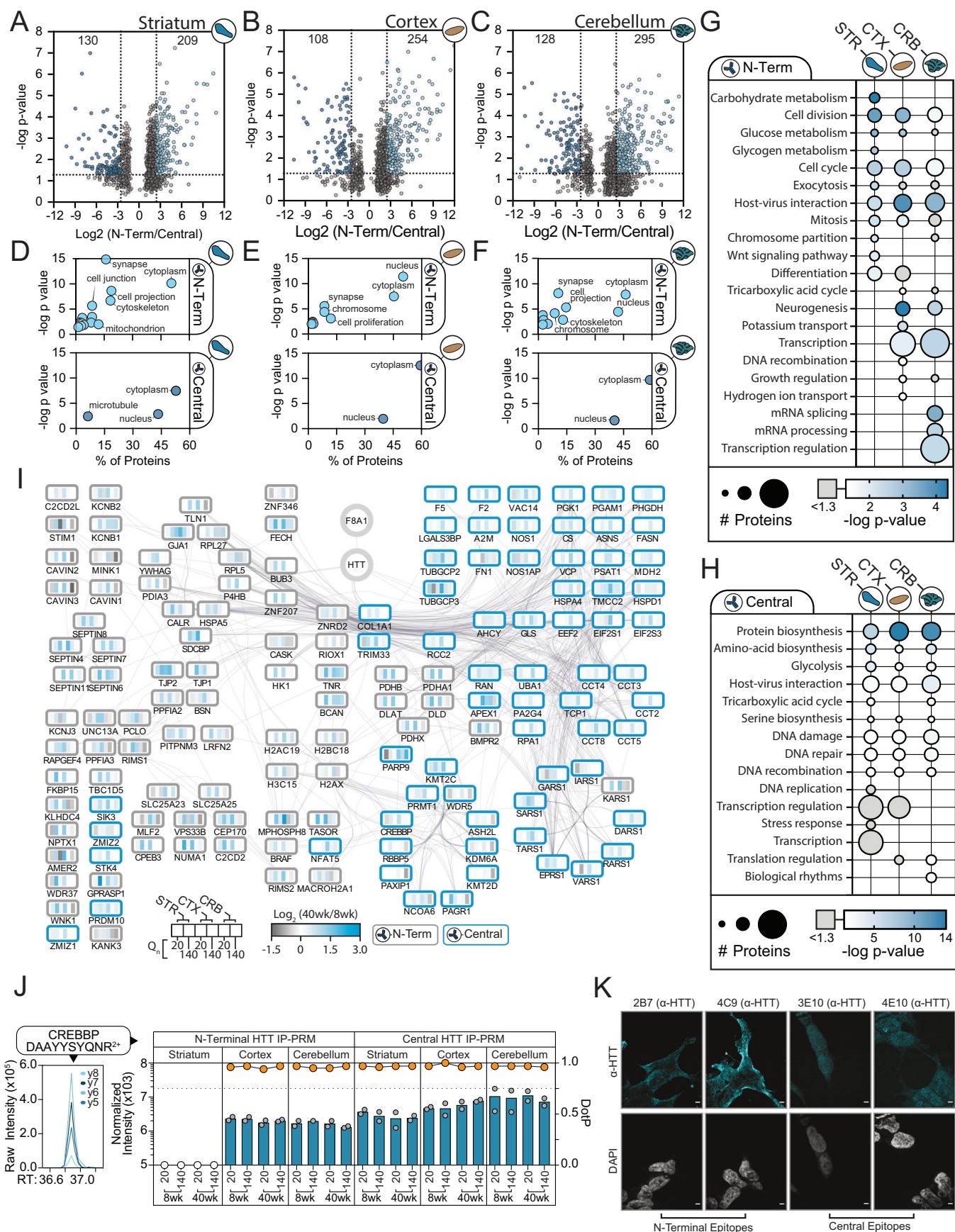

**Figure 2.   Antibody epitope target drives distinct HTT interaction signatures between tissues of the brain.**

(A–C) HTT PPI (HIP) abundance ratio of N-terminus versus Central antibody IPs from the striatum (A), cortex (B), and cerebellum (C). PPI ratios of N-terminal versus Central abundance for each genotype-age condition were calculated, and the ratios with the greatest absolute value within each tissue were plotted. HIP ratios with |log$_2$| >2.5 were considered preferentially captured. *P* values were calculated by unpaired *t* tests (*n* = 3–4 per group). (D–F) Subcellular localization ontology enrichment analysis was performed for HIPs captured in N-terminal (top) or Central (bottom) IPs in the striatum (D), cortex (E), and cerebellum (F). (G, H) Pathway enrichment analysis for HIPs captured in N-terminal (G) or Central (H) IPs in the striatum, cortex, and cerebellum was performed using DAVID (https://davidbioinformatics.nih.gov/). The top ten pathways were collated for each tissue type and the number of annotated proteins and enrichment significance (*P* value, Fisher's Exact test) are displayed for each pathway. (I) The subset of HIPs identified in all three tissues was assembled into a STRING functional network. Nodes are color-coded to represent the HIP abundance age ratio (40 weeks/8 weeks) for each genotype (Q20 and Q140) in the striatum, cortex, and cerebellum to visualize age-dependent interactions. Node border color indicates whether the HIP was captured with the N-term (gray) or Central (blue) antibodies. (J) Targeted MS/MS (parallel reaction monitoring) quantification of CREBBP abundance in HTT IPs, as indicated. For each condition, the median normalized intensities (left axis, bars, *n* = 2 biological replicates) and chromatogram peak DotP score (right axis, line graph) for the CREBBP peptide were plotted. A DotP score >0.75 indicates a high-confidence assignment (orange circles). (K) Immunofluorescence microscopy of HTT staining (cyan) with epitope-targeted antibodies in BE(2)-C neuroblastoma cells. Nuclei were visualized with DAPI (grayscale). Scale bar = 10 µm. Source data are available online for this figure.

our complete HIP results (Dataset EV2) for functional over-representation of protein complexes annotated in the CORUM database. Dozens of protein complexes were statistically enriched, including the Mediator of RNAPII transcription complex, the ribosome, and several chromatin remodeling complexes (Fig. 3A). Functional annotation of the proteins associated with these complexes showed high enrichment for transcription regulation, followed by protein biosynthesis (Fig. 3B). We next asked if functionally enriched protein complexes displayed an HTT epitope or genotype linked characteristic. Quantifying the abundance of the ~60 ribosomal proteins that co-isolated with HTT showed a clear preference in the N-terminus targeted IP condition that was especially pronounced in cerebellum (Fig. 3C, top). In general, HTT-Q140 showed a somewhat decreased association with the ribosome from 8 weeks to 40 weeks compared to HTT-Q20 (Fig. 3C, bottom and 3G). Recently, it was shown that HTT mutation could induce stalling of translation elongation in mouse HD striatal neuronal cells, which suggests a functional association between HTT and protein biosynthesis (Eshraghi et al, 2021). Another HTT-enriched interaction was with several subtypes of voltage-gated calcium ion channel proteins. These proteins showed a strong preference for the HTT N-terminus in the striatal tissue that was less consistent in the cortex and cerebellum (Fig. 3D, top). Examination of age-linked polyQ dependence showed a striking mHTT-dependent increase in the striatum that was less prevalent in the cortex and absent in the cerebellum (Fig. 3D, bottom, 3H). This effect was particularly pronounced for the HTT interaction with CACNA1A, the alpha-1A subunit of the P/Q-type voltage-gated calcium channel. Interestingly, the CACNA1A gene is subject to CAG repeat expansion, where it causes spinocerebellar ataxia type 6 (Zhuchenko et al, 1997). To date, numerous pathogenic variants in the CACNA1A gene have been documented, with links to other ataxias and neurodevelopmental conditions (Lipman et al, 2022).

We observed a prevalent interaction between HTT and chromatin remodeling complexes. Specifically, we found that the BAF complex (the SWI/SNF complex) was preferentially enriched by the Central HTT IP in the striatum, while being preferentially enriched in the cortex and cerebellum in the N-terminal IP (Fig. 3E). The majority of BAF complex members lost their interaction overtime in the Q140 versus the Q20 mice in the striatum and cortex tissues, which was not observed in the cerebellum (Fig. 3F). Notably, HTT's impact on cellular chromatin

remodeling and transcription regulation is a topic of interest, as HTT N-terminal fragments can drive inclusion formation in the nucleus and affect RNA metabolism and DNA handling (DiFiglia et al, 1997; Smith et al, 2023; Landles et al, 2020). However, the epitopes targeted by the Central IP are C-terminal to the nuclear HTT fragment species, which raises the possibility that full-length HTT may interact with the BAF complex in the nucleus. Alternatively, the cytosolic interaction of typically nuclear proteins with HTT could impact the function of these complexes. Overall, we observed a tendency to isolate interacting proteins that reflect the whole or the majority of components of multi-subunit protein complexes. As subunits of these complexes often displayed concerted disruption due to polyQ and/or within specific tissues, the primary functions of these complexes could be differentially impacted by mHTT toxicity among tissues.

## mHTT interacts with the Mediator complex in multiple tissues but is differentially regulated and alters Mediator subcellular localization

Based on our functional enrichment analysis of CORUM complexes, we found that HTT was associated with almost every member of the canonical 26 subunit Mediator complex, which coordinates long-range transcription enhancer element function and transcription initiation via RNA polymerase II (Pol II) (Richter et al, 2022; Kim et al, 1994). While the HTT-Mediator interaction was detected across tissues, the ability to capture Mediator from different tissues was epitope-dependent. Specifically, Mediator was captured with higher levels by the N-terminal IP in the cortex and cerebellum, and by the Central IP in the striatum (Fig. 4A). This observation is noteworthy given our finding that epitope-enriched HIPs are functionally distinct (Fig. 2). An exception to this pattern was the HTT interaction with Med15, which was enhanced in the N-terminal IP in the striatum. Like HTT, Med15 is a glutamine-rich protein. Our results show that the association of mHTT with Mediator increased in the striatum and decreased in the cerebellum (Fig. 4B). Next, we investigated which Mediator subunits could be at the interaction interface with HTT. We hypothesized that interacting subunits with the greatest IP abundance would be likely candidates. To permit inter-subunit comparisons, subunit abundances were normalized by their respective theoretical number of tryptic peptides (Schwanhüusser et al, 2011). Subunit abundances were not uniform across the Mediator complex; rather, we found

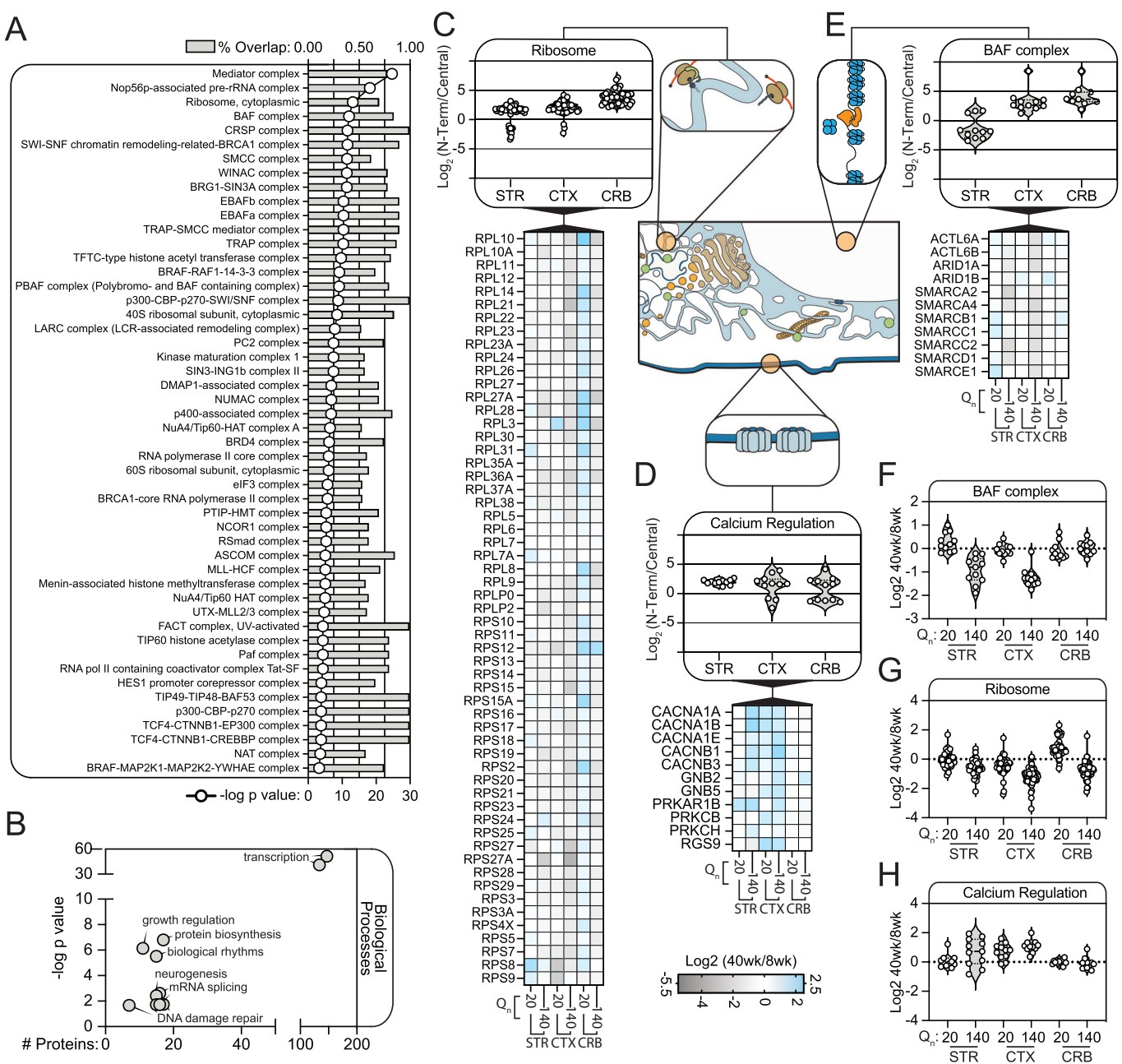

**Figure 3. HTT interacts with large macromolecular protein complexes.**

(A) CORUM complex analysis for HIPs from the striatum, cortex, and cerebellum. Complexes missing ≥50% of members were removed. (B) Cellular processes gene ontology enrichment analysis for proteins within assigned CORUM complexes from (A). (C–E) HIP abundance ratios for subunits annotated to selected CORUM complexes, ribosome (C), calcium channels (D), and BAF complex (E). Top plots are summarized HIP abundance ratios, N-terminus vs. Central, in the striatum, cortex, and cerebellum. The bottom plots are heatmaps of 40 weeks vs. 8 weeks HIP abundance ratios for each genotype and tissue. Comparison of HIP ratios between Q140 and Q20 indicate polyQ-dependent changes in IP abundances. (F–H). Heatmap ratios (C–E) are plotted for each genotype and tissue for members of the BAF complex (F), ribosome (G), and calcium channels (H).

enrichment for the head domain of Mediator, and the proteins that form the interface between the head and tail domain—Med15 and Med27 (Fig. 4C). Conversely, subunits that form the middle and distal tail domains were relatively lower in abundance or not detected. These observations suggest that HTT interacts with a few specific subunits of Mediator, with the remainder of the complex being enriched via bridging.

Given that we observed different interaction patterns between Mediator and HTT in epitope-targeted IPs, we sought to further refine and validate this interaction. We performed IP-MS using individual antibodies rather than a combination of antibodies to determine whether the interaction could be recapitulated by multiple, independent isolations. From the striatum of Htt-Q20 mice, Mediator was enriched with HTT in three of five IP

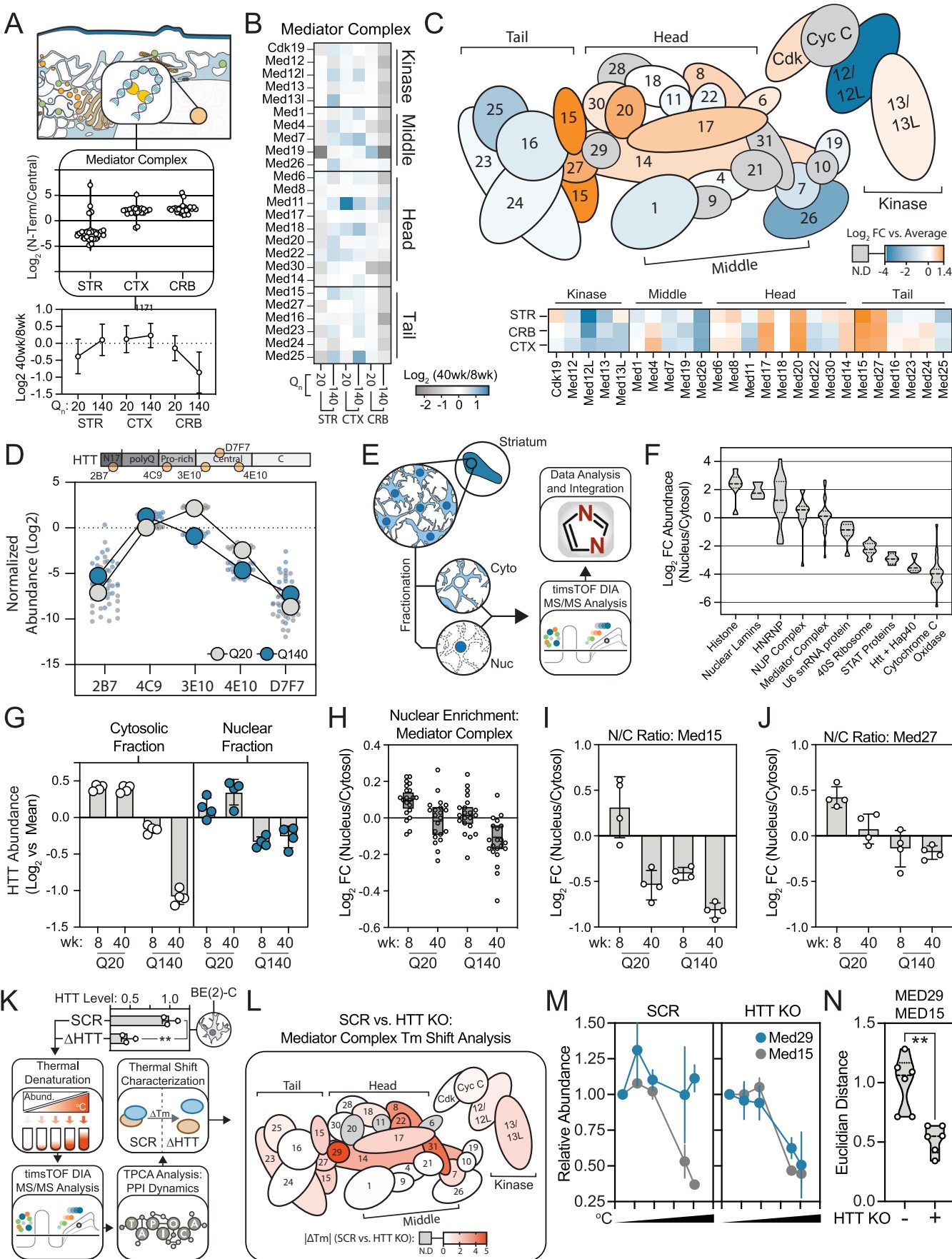

**Figure 4. HTT interacts with the Mediator Complex.**

(A) HTT interaction abundance with Mediator complex subunits, plotted as HTT antibody (N-terminus vs. Central HIP abundances) (top) or age (40 weeks vs. 8 weeks) (bottom) ratios for subunits quantified in each genotype (Q20 vs. Q140 HTT) in the striatum, cortex, and cerebellum ($n = 3$–5 biological replicates). (B) Heatmap of HIP age ratio changes between genotypes and tissues. (C) Estimated inter-subunit IP abundances, relative to the average Mediator subunit IP abundance. Mediator HIP abundances were scaled by the number of theoretical tryptic peptides. Illustration of a simplified structural model of the Mediator complex with subunits colored by their abundance in the striatum IPs (top). A heatmap representation comparing scaled Mediator subunit IP abundances across tissues (bottom). (D) Orthogonal validation of Mediator subunit HTT interactions by IP-MS with single HTT antibodies, as indicated on the HTT domain map. Shown are the aggregated mean subunit abundances for the Q20 (gray) and Q140 (blue) genotypes at 40 weeks, with abundances normalized by the average within each genotype for each protein. (E) Workflow diagram for nuclear-cytosolic fractionation of mouse striatum from HTT-Q20 and Q140 at 8 and 40 weeks. Following fractionation of frozen mouse striatum, whole protein quantification was performed by nanoLC-MS/MS on a timsTOF Ultra mass spectrometer operated in data-independent acquisition mode and analyzed by DIA-NN. (F) Aggregated protein ratios (nucleus/cytoplasm, N/C) for selected proteins/complexes that have preferential localization to the nucleus or cytoplasm. For each protein within the plotted groups, N/C ratios were calculated as an average of the four genotype-age conditions and four replicates, except for Htt and Hap40, where individual N/C ratio values were used ($n = 16$ values). The N/C protein ratios were plotted as distributions. The number of total proteins represented within each group, from left to right, were $n = 7, 3, 18, 18, 23, 8, 32, 4, 2,$ and 24. (G) Mean-normalized $\log_2$ HTT abundances within each cell fraction across genotypes and ages (mean ± SD, $n = 4$ biological replicates). (H–J) Aggregate subunit ratios (nucleus/cytosol) by genotype and age for all mediator subunits (H), Med15 (I), and Med27 (J) (mean ± SD, $n = 4$ biological replicates). (K) An HTT KO model and scrambled KO control (SCR) were established using CRISPR in BE(2)-C cells, which were then subjected to thermal profiling. The abundance of HTT in cell populations from SCR or HTT KO treatments was quantified by label-free MS quantification ($n = 3$ biological replicates, unpaired $t$ test, **$P = 0.002$). (L) The absolute value of $\Delta$Tm shifts between SCR and HTT KO BE(2)-C cells for Mediator subunits was determined with thermal profiling and mapped to the Mediator structure diagram. (M) Thermal profiles for Med15 and Med29 in SCR and HTT KO cells ($n = 3$ biological replicates per point, mean ± 95% confidence interval). (N) The pairwise Euclidean distances between the Med15 and Med29 thermal solubility curves were calculated between each of three biological replicates (for a total of six comparisons) in SCR and HTT KO cells. $P$ values were calculated by an unpaired $t$ test (**$P = 0.0013$).

conditions (Fig. 4D), confirming the robustness of this association. Mediator was most abundant in the IP using the 3E10 (Central) antibody, which targets a 200 amino acid epitope that is C-terminal to the HTT polyQ and proline-rich regions. Strong enrichment of Mediator was also observed following IP with the 4C9 antibody, which targets the proline-rich region C-terminal to the polyQ domain. This pattern was reversed for the polyQ expanded HTT (Q140), which showed a slight preference for the 4C9 antibody. Interestingly, for both HTT-Q20 and Q140, the 2B7 antibody that targeted the extreme N-terminus only enriched for a subset of mediator subunits. One possibility is that Mediator may interact with the N-terminus of HTT, thereby obscuring a subset of the bait from immunocapture. Overall, these findings reveal a reproducible HTT-Mediator interaction that is impacted by pathogenic polyQ expansion.

The HTT interaction with the Mediator complex is only one example of associations we identified with proteins or protein complexes with nuclear annotation. In the striatum, associations with nuclear proteins were more prevalent when capturing via the Central epitopes. Supporting this observation, immunofluorescence microscopy of HTT staining in BE(2)-C neuroblastoma cells showed a perinuclear enrichment localization when using antibodies against the Central epitopes (Fig. 2K). In the literature, HTT is reported as primarily cytosolic, while nuclear-localized HTT is thought to likely reflect N-terminal fragments that, as noted above, have been implicated in HD pathobiology (DiFiglia et al, 1997; Landles et al, 2020; Smith et al, 2023). However, the 3E10 and 4E10 (Central) epitopes are not present in N-terminal HTT fragments. This raises the question if the nuclear-annotated proteins we co-isolated with HTT using the Central antibodies may represent cytosolic associations with multilocalized or mislocalized proteins. To address this question, we performed cellular fractionation of striatal tissue to generate nuclear and cytoplasm enriched fractions, followed by DIA quantitative proteomics analysis (Fig. 4E). As expected, HTT was found predominantly in the cytosolic fraction, but was also detectable at low levels in the nucleus enriched fraction, which showed ≥ four-fold enrichment of histone and other nuclear markers (Fig. 4F). Quantification of HTT relative

abundance within the cytoplasmic fraction recapitulated the trend of decreased HTT-Q140 in aged (40 weeks) striatum observed in whole cell lysates (Figs. 4G vs 1C). The nuclear-enriched fraction also showed a decrease in HTT-Q140, but aged mice did not demonstrate the same pronounced decrease in HTT levels as observed in the cytosol. These observations suggest that a distinct pool of HTT-Q140 may be associated with the nucleus as HD progresses.

When assessing Mediator subcellular localization, we found roughly equivalent distribution between the nuclear and cytoplasmic fractions (Fig. 4H). This observation suggests a degree of nuclear-cytoplasmic shuttling, which is consistent with the biological function of Mediator. Mediator was predominantly enriched in the nucleus of Htt-Q20 8-week mice, a localization that shifted to become equivalent between the two fractions as the mice aged. HD mice (Q140) also displayed an apparent shift in Mediator enrichment toward the cytosolic fraction; however, HTT-Q140 8 weeks mice were phenotypically similar to aged HTT-Q20 mice. These data suggest that polyQ expanded HTT may bias the subcellar localization of the Mediator complex. Our IP-MS experiments revealed a particular enrichment for the interaction between HTT-Q140 and the Head-Tail interface Mediator subunits, Med15 and Med27, while at the proteome level these two subunits showed a Q140-dependent enrichment in the cytosol (Fig. 4I,J). Therefore, a (dys)functional relationship between pathogenic HTT and Mediator may impact Mediator's cellular distribution and activity.

Mediator is a multi-subunit protein complex with its activity regulated by the dynamic assembly and disassembly of regulatory accessory subunits (Richter et al, 2022). To understand if HTT interaction with the Mediator complex influences the latter's composition or function, we coupled genetic modulation of HTT expression in a neuroblastoma cell model, BE(2)-C cells, with a thermal proximity coaggregation assay (TPCA) (Fig. 4K). TPCA relies upon the principle that proteins that interact tend to display similar aggregation dynamics upon progressive heat denaturation (Tan et al, 2018; Hashimoto et al, 2020). When coupled with mass spectrometry to compare the thermal profiles of proteins in

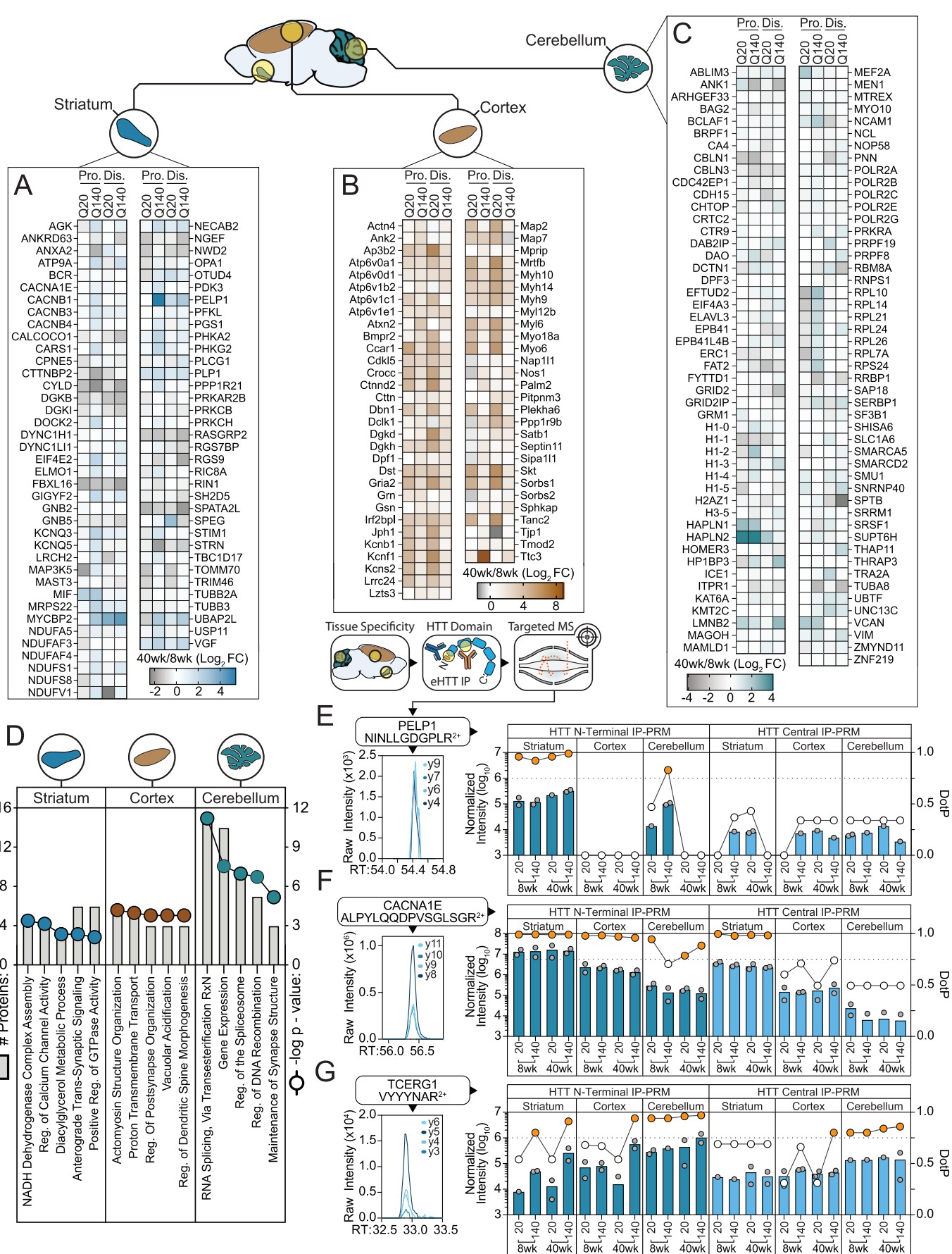

**Figure 5. Tissue-enriched eHTT protein-protein Interactions.**

(A–C) Heatmaps of HIPs enriched by $\log_2 >2$ in the (A) striatum, (B) cortex, or (C) cerebellum versus the other two tissues. (D) Gene ontology enrichment analysis for the tissue-enriched proteins in (A–C) was performed using Panther (https://www.pantherdb.org/). Bars are the number of identified proteins for each term (left axis) and line graph is the $-\log_{10}(P$ value) (right axis), calculated by a Fisher's Exact test. (E–G) Targeted MS/MS (parallel reaction monitoring) quantification of representative peptides from (E) PELP1, (F) CACNA1E, and (G) TCERG1 in HTT IP sample conditions as indicated. For each condition, the median normalized peptide intensities (left axis, bars, $n = 2$ biological replicates) and maximum chromatogram peak DotP scores (right axis, line graph) were plotted. A DotP score >0.75 indicates a high-confidence assignment (orange circles).

different conditions, TPCA can reveal changes in melting temperature that are indicative of differential interactions between a protein and its cellular environment, such as the dynamic assembly of protein complexes. We interrogated the melting signatures of Mediator complex proteins in HTT KO versus scrambled (SCR) control BE(2)-C cells, using the TAPIOCA computation tool (Reed et al, 2024) for thermal profiling data analysis. Changes in melting temperature ($\Delta$Tm) were mapped to the structure of Mediator, revealing a "hotspot" at the interface of the tail and body sub-complexes (Fig. 4L). Specifically, Med29, Med31, Med8, and Med 22 had the most evident Tm shifts, while other complex members also displayed changes in Tm, suggesting a broader change in the complex. When assessing the melting curves of Med29 and Med15, we noted that the Tm change was accompanied by a shift in the profile of Med29, which became more similar to the Med15 profile in HTT KO cells (Fig. 4M). Indeed, a significant decrease in the Euclidian distance between the Med 29 and Med15 solubility curves was found in HTT KO cells (Fig. 4N), indicating increased curve similarity and, thus, potential enhanced interaction. Therefore, the association of HTT may destabilize this interface of the complex, as these subunits become better associated in the absence of HTT. Curiously, from our endogenous HTT-IP MS experiments, we did not co-isolate Med29 with (m)HTT. Since we hypothesized that HTT is binding to Med15 or Med27, it is possible that HTT hinders Med29 binding. Alternatively, Med29 may be a transient interaction and not easily co-isolated with HTT by IP. Overall, our results demonstrate a specific interaction between HTT and the Mediator complex that affects the composition and cellular distribution of this complex.

## Tissue-enriched interactomes point to tissue-specific (m)HTT associations and functions

Although HD impacts several brain regions, the striatum tissue is profoundly affected by HD progression. Advanced HD is marked by nearly total striatal atrophy and neuron loss with associated decreases in cortical white matter (Waldvogel et al, 2014). Manifestation of motor symptom phenotypes associated with HD progression has also been linked to loss of cerebellar Purkinje cells in humans (Rüb et al, 2013; Jeste et al, 1984). However, this loss in the human disease is not recapitulated in mouse HD models until an advanced stage of HD-like pathology (Turmaine et al, 2000; Dougherty et al, 2013). In our primary interactome analysis (Figs. 1 and 2), we observed that epitope-enriched interactions tended to be associated with certain functional classes in specific tissues. In this context, the striatal landscape of HIPs had unique epitope-dependent features. For example, the N-terminal IP of HTT from the striatum was underrepresented in interactions with nuclear annotations but were over-represented in the cortex and

cerebellum. This was corroborated by analysis of CORUM complexes (Fig. 3), which showed that capture of transcription-related complexes (e.g., BAF and Mediator) from the striatum occurred when isolating HTT with the Central IP. Taken together, the composition and topology of HIPs and/or their interaction interfaces are important aspects of HTT biology and likely support tissue-specific functions. Therefore, we pursued explicit classification of tissue-specific HTT interactions.

Focusing first on the striatum, interactions were filtered for those enriched over four-fold versus the cortex and cerebellum (Fig. 5A). The resulting ~70 proteins were functionally interconnected, with multiple members contributing to calcium ion transport, mitochondrial organization, and NADH dehydrogenase activity (Fig. 5D). Specifically, we again observed enrichment for several members of the calcium ion channels (CACNA1E, CACNB1, etc.) that were represented in the N-terminal IP condition. Calcium dyshomeostasis is a unifying feature of HD models (Giacomello et al, 2013) and mutant HTT contributes to calcium-linked cell death via membrane permeability of the mitochondria (Choo et al, 2004), supporting that our striatum-enriched HIPs correlate to processes linked to HD pathobiology. The same analysis in the cortex enriched for ~50 proteins (Fig. 5B) that were linked to trafficking and lysosome acidification (Fig. 5D). The ~90 cerebellum enriched targets (Fig. 5C) were representative of chromatin handling proteins that are linked to splicing, DNA repair, and gene expression (Fig. 5D). These data imply a strong tissue preference for HTT interactions and provide factors that can mediate tissue-specific cytoprotection versus pathology.

To validate these interactions, we performed targeted mass spectrometry experiments on HTT immunoisolates across all polyQ-age, domain, and tissues conditions (Fig. 5E–G; Dataset EV3). One notable polyQ-dependent candidate in the striatum was the Proline-, glutamic acid-, and leucine-rich protein 1 (PELP1), which has diverse cellular roles in signaling, rRNA processing, and transcription. Targeted quantitative MS validation supported the increase in PELP1 interaction with age, as well as potentiation with mHTT in the striatum, while this interaction was not detected in the cortex (Fig. 5E). We also confirmed that the putative striatum-enriched interaction, CACNA1E, a subunit of voltage-dependent R-type calcium channels, was on average nearly 10-fold and >50-fold higher in the striatum versus the cortex and cerebellum, respectively (Fig. 5F). Finally, we targeted the putative interacting partner, transcription elongation regulator 1 (TCERG1). In our global interactome analysis, TCERG1 was scored as a positive interaction in the cerebellum of Htt-Q140 mice but was filtered out due to subthreshold spectral counts. TCERG1 has been reported as an HTT interaction in yeast two-hybrid studies (Holbert et al, 2001) but had not been previously confirmed as interacting in mammalian models. Using targeted MS, we were able to

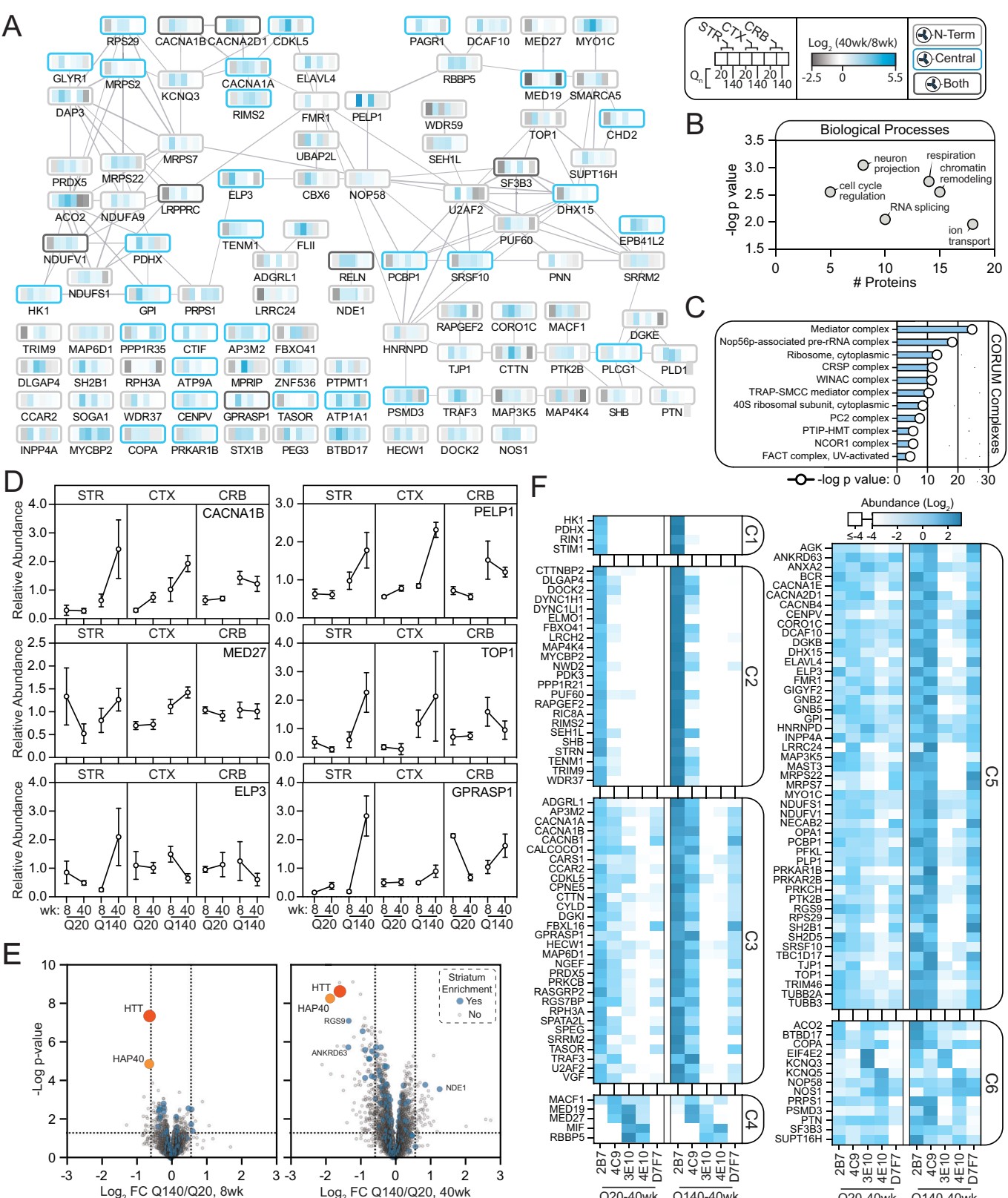

**Figure 6.  Striatum-enriched and polyQ-dependent HIP network.**

(A) STRING-based functional network of HIPs that increased in abundance >2.5-fold in Q140 vs. Q20 mice (comparing 40 weeks/8 weeks ratio) and had a striatum tissue enrichment of >two-fold. (B) GO biological process enrichment analysis for HIPs in (A) was performed using DAVID (https://davidbioinformatics.nih.gov/) with *P* values calculated by a Fisher's exact test. (C) Over-represented CORUM complexes (from Fig. 3) with at least one subunit that corresponds to a striatum enriched and polyQ-dependent HIP from (A) were curated by EnrichR (https://maayanlab.cloud/Enrichr/). The significance of enrichment was calculated by a Fisher's Exact test and corrected for multiple hypothesis testing by the Benjamini-Hochberg method. (D) Representative abundances ($n = 3$–4 biological replicates, mean ± SD) for HIPs identified in (A). (E) Volcano plots of polyQ-dependent differential abundance quantified by mass spectrometry from the striatal proteomes of 8 and 40 weeks mice ($n = 3$–4 biological replicates, mean ± SD). *P* values for each protein were calculated using unpaired t-tests, using a threshold of 0.05. Blue-colored nodes highlight striatum-enriched HIPs. (F) Orthogonal HTT IP-MS validation of striatum enriched HIPs from Figs. 5 and 6 using single HTT antibodies as indicated. Interactions were clustered by the k-means algorithm.

confidently measure TCERG1 interaction in the cerebellum, which exhibited polyQ-dependence at 40 weeks of age (Fig. 5G). Overall, our quantitative analysis of tissue-enriched HIPs provides a tractable set of enriched targets whose activities could underpin the incongruence between mHTT expression and tissue suscept-ibility to HD pathogenesis.

## mHTT modulation of striatum-enriched interactions highlights candidates proximal to the disease

We next sought to determine if a subset of HTT interacting proteins were differentially impacted by polyQ expansion and HD progression in the striatum. This analysis was independent of tissue enrichment, so that we could include HTT interactions shared in all tissues but impacted by mHTT only in the striatum. Given that there is no effective therapeutic intervention for HD onset or disease progression, we focused our analysis on gain-of-interaction phenotypes, as these may represent future candidates for ther-apeutic disruption. We selected HTT interactions that increased in abundance from 8 to 40 weeks in Q140 versus Q20 control mice by >five-fold in the striatum, but not in the cortex or cerebellum. The resulting 104 proteins partially overlapped with the subset of striatum-enriched interactions (Fig. 6A). For example, HTT interaction with PELP1 was both enriched in the striatum and with polyQ expansion. Again, ion transport emerged as a significantly enriched functional category when filtering for polyQ modulated interactions (Fig. 6B). Interactions with calcium channel proteins, such as CACNA1B and CACNA1A were significantly and disproportionately enhanced in the Q140 mice in the striatum and, to a lesser extent, the cortex, while those linked to chromatin handling and RNA metabolism were also dysregulated in the striatum, including members of the Mediator complex, Med27 and Med19 (Fig. 6C,D).

To rule out that changes in interactions were primarily driven by HD-dependent proteome alterations, we performed total proteome quantification matching our IP-MS experimental conditions. The overwhelming majority of HTT interacting proteins from the striatal polyQ network were not differentially regulated at the proteome abundance level (Fig. 6E), similar to our prior observations in a FLAG-HTTQ140 mouse model (Greco et al, 2022). Finally, we further confirmed these interactions by independent IP-MS experiments using single antibodies. As visualized by hierarchical clustering analysis, the majority were detected by more than one antibody, but as predicted, distinct clustering remained driven by N-terminal (2B7/4C9) (C1 & C2) versus Central (3E10/4E10) (C4) antibodies (Fig. 6F). Overall, combining striatum-enriched and polyQ responsive HTT

interactions in the striatum resulted in 165 high-priority, reproducible HTT interacting proteins.

## Curation of HD genetic modifiers reveals a functional network of striatum- and mHTT-enriched interactions that contribute to the regulation of calcium levels by HTT

To place our results into the broader context of the HIP knowl-edgebase, we integrated our tissue consensus and striatum-linked HIPs with previously published HTT interactions annotated in the HINT database (Aaronson et al, 2021). More than 2/3 of the HIPs from our primary tissue consensus analysis were annotated in HINT, leaving 399 HIPs as unreported candidates (Fig. 7A). Compared to the collection of HIPs shared with HINT, the unreported HIPs were enriched in biological functions that are of evolving interest in HD pathobiology—e.g., RNA metabolism and transcription (Fig. 7A). Comparison of the subset of 165 high-priority striatum enriched targets (Fig. 6) showed that the majority (55%) overlapped with HINT annotated HIPs (Fig. 7B). At the functional level, these common HIPs reflect cellular processes in RNA transcription and chromatin dynamics. A subset of the high-priority striatum-enriched proteins (45%, $n = 75$) were not previously reported (Fig. 7B). These proteins functionally linked HTT to ion homeostasis and related synaptic membrane potential. Cross-referencing our striatum-enriched HIPs with murine phe-notypes uncovered enrichment for proteins, specifically calcium channel subunits that are associated with locomotive dysfunction (Fig. EV2A). These locomotive annotated interactions had on average increased polyQ-dependent association with HTT, which was not observed in the cortex or cerebellum (Fig. EV2A, bottom). Overall, the bioinformatics analysis of our tissue consensus HIPs provided validating evidence for ~20% of extant HIP candidates, while also uncovering unique targets that reflected biological processes that are underrepresented in the HINT database.

The need to advance HD therapeutics brings the disease relevance of HIPs into primary focus. Towards this goal, we computationally evaluated which of our striatum-linked HIPs are modifiers of HD at the genetic level. We leveraged the curated HD dataset containing perturbation studies on HDinHD.org. We focused on perturbations at the genetic level and removed those that had no change in the outcome (see "Methods"). In total, we obtained ~6750 unique perturbations, which mapped to 3565 genes with human orthologs that we will refer to as HD genetic modifiers. Fifty-four of the 165 striatum-linked HIPs were annotated as genetic HD modifiers, representing a total of 81 independent experimental results (Fig. 7C). About half the results were obtained from rodent HD models, while 14 were from human models, 12 of

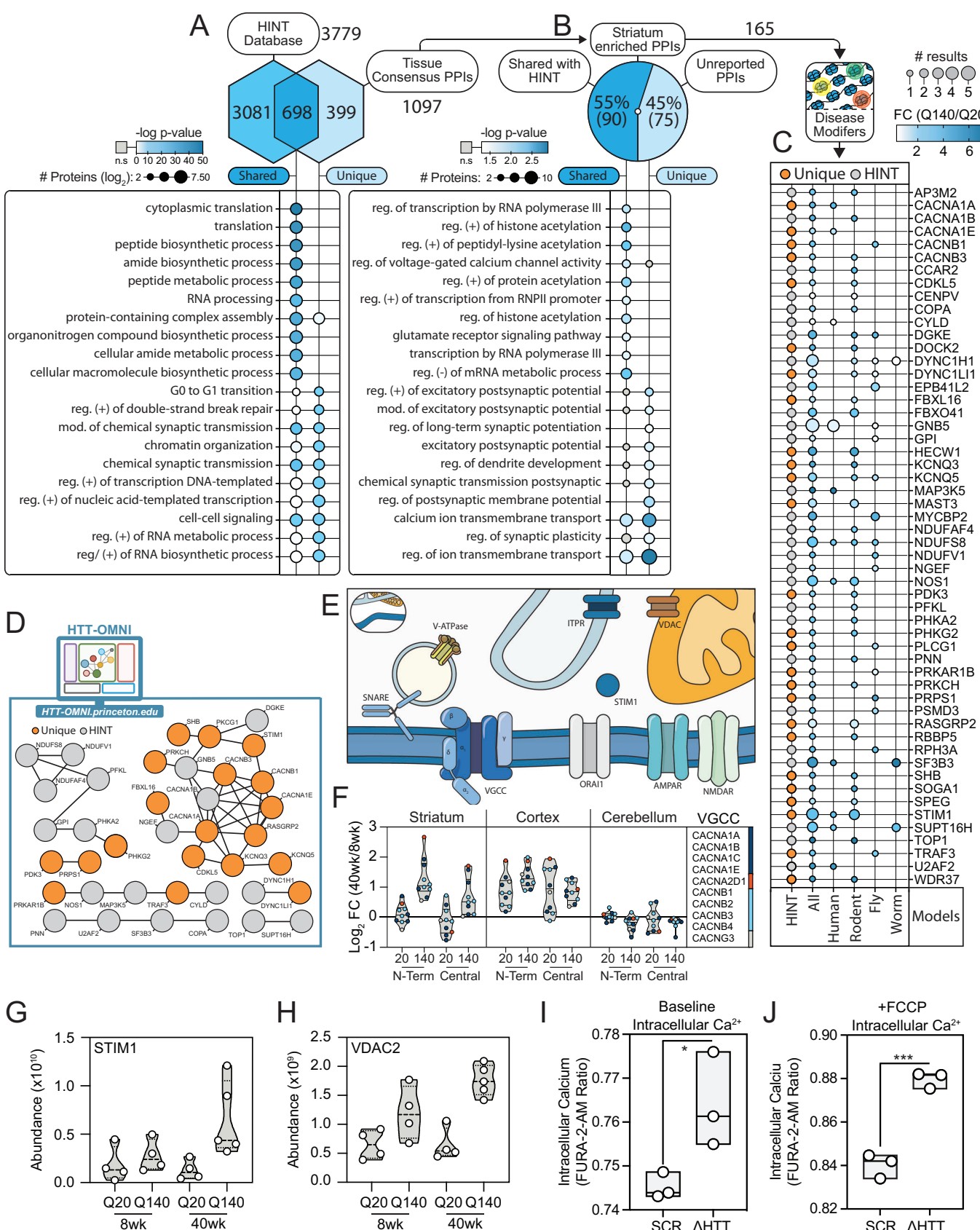

**Figure 7. HTT PPIs that are genetic disease modifiers impact calcium homeostasis.**

(A) Overlap of HIPs identified in this study (tissue consensus PPIs) with HIPs curated in the HINT database. Over-represented biological processes for shared ($n = 698$) and unique ($n = 399$) proteins were determined by DAVID (https://davidbioinformatics.nih.gov/) and visualized by dot plots. Enrichment *P* values were calculated by Fisher's Exact test. (B) The proportion of striatum-enriched HIPs identified in this study ($N = 165$) that have been curated in the HINT database versus previously unreported. Over-represented biological processes for shared ($n = 90$) and unique ($n = 75$) proteins were determined by DAVID (https://davidbioinformatics.nih.gov/) and visualized by dot plots. Enrichment *P* values were calculated by Fisher's Exact test. (C) Striatum-enriched HIPs that are known genetic disease modifiers. Genetic modifiers were curated from the Perturbations database available from HDinHD.org (see Methods). Annotated HIPs were visualized in the dot plot as a function of the genetic HD model (All, Human, Rodent, Fly, and Worm), number of observations (node size), and striatum polyQ-dependence (node fill color). (D) STRING-based interaction network of striatum-enriched, HIP genetic modifiers were assembled using HTT-OMNI. (E) Illustration of calcium regulatory proteins at the plasma membrane, endoplasmic reticulum, and mitochondria. (F) Violin plots with age abundance ratios for voltage-gated calcium channel subunits ($n = 10$ proteins, mean ± SD) showing the impact of polyQ HTT among striatum, cortex, and cerebellum tissues for N-terminal and Central antibodies. (G, H) Violin plots of normalized MS abundance for STIM1 (G) and VDAC2 (H) in the striatum following HTT-IP (N-terminal antibodies). The mean and 95% confidence interval is illustrated by thick and thin dashed lines, respectively ($n = 4$–5 biological replicates). (I, J) Relative intracellular calcium level quantified by FURA-2-AM in scrambled control (SCR) and HTT Crispr knockout (ΔHTT) BE(2)-C neuroblastoma cells under baseline (I) and FCCP treatment (J) to stimulate oxidative mitochondrial stress and calcium flux. Boxes represent minima and maxima of data points with an internal line indicating the median ($n = 3$ biological replicates, unpaired *t* tests, *$P = 0.0431$, ***$P = 0.0005$).

which were identified in cell models and 2 (CACNA1A and MAP3K5) in human studies. These genetic modifiers were also largely polyQ-dependent, 65% being perturbed by mHTT by ±two-fold or greater. To visualize potential functional and multi-omic relationships, we analyzed the HIP modifiers with our web-based platform, HTT-OMNI (Kennedy et al, 2022) (Fig. 7D). Most HIP modifiers had at least one known functional relationship in STRING. Notably, the highest interconnectivity reflected HIPs involved in calcium homeostasis and ion transport (Fig. 7D), echoing other functional enrichments identified in this study. Using HTT-OMNI's multiomics visualization tools, we observed that HIP modifiers had an average RNA fractional expression that was greatest in MSNs and interneuron cell types (Fig. EV2B), as quantified from single cell transcriptomics of wild-type mouse striatum (Gokce et al, 2016). From single nuclei transcriptomics (Malaiya et al, 2021), a trend towards increased Q175-dependent RNA expression of these HIP modifiers was observed; however, the effect sizes were small (Fig. EV2C). This observation is consistent with prior studies that documented minor polyQ-dependent correlations between interactomes and proteomes/transcriptomes (Shirasaki et al, 2012; Greco et al, 2022; Federspiel et al, 2019). Overall, this reinforces the selective impact that expanded polyQ likely has on HTT-containing protein complexes, while also showing promise for therapeutic intervention given our identification of known disease-relevant HIPs.

Analysis of striatum-enriched HTT PPIs uncovered a cohort of proteins associated with calcium homeostasis, including voltage-gated calcium channels (P/Q-, R-, and L-types), and a regulator of store-operated $Ca^{2+}$ channels (STIM1) (Fig. 7E). Not only were these interactions enriched in the striatum but they were modulated by mHTT (Fig. 7F,G). For example, ten subunits that assemble voltage-gated calcium channels selectively increased their interaction with HTT Q140 in the striatum (Fig. 7F). STIM1, an ER calcium sensor, also increased in association with mHTT, but only in 40 week old mice (Fig. 7G). We noted that inter-organelle ion transport was an enriched feature of HTT-dependent interactions, including VDAC2, which is a mitochondrial ion channel and ITPR1, which is an inositol triphosphate ($IP_3$) receptor that regulates intracellular calcium. ITPR1 interaction levels did not show a consistent mHTT-dependent effect and had higher biological variability (Fig. EV2D), while the interaction levels of VDAC2 with mHTT were consistently increased relative to controls (Fig. 7H). Given these interactions, we next asked if

HTT contributes to calcium homeostasis or cell stress-induced calcium regulation. We re-visited our HTT KO model in BE(2)-C and measured intracellular calcium using the fluorometric reporter, FURA-2-AM. Upon HTT KO, relative intracellular calcium was significantly increased compared to SCR KO controls (Fig. 7I). Next, we asked if HTT KO altered calcium flux due to mitochondrial oxidative stress (Luo et al, 1997). We treated SCR and HTT-KO BE(2)-C cells with the electron transport chain decoupler, FCCP, to trigger calcium flux from mitochondrial stores (Jadiya et al, 2019). We observed significantly increased intracellular calcium in HTT-deficient cells (Fig. 7J), which could indicate the high sensitivity of these cells to death linked to calcium dyshomeostasis (Matuz-Mares et al, 2022). These findings support a role for HTT in regulating calcium flux, which becomes impaired in HD, as reflected by aberrant striatum-enriched, mHTT-dependent protein interactions.

## Identification of HTT interactions that impact neuronal function and phenotypes in an HD fly model

Among our identified 165 striatum-linked HIPs (Fig. 7B), the majority ($n = 111$) have not been previously reported as genetic HD modifiers (Dataset EV4). These HIPs were annotated to several cellular processes, such as synaptic transmission, consistent with functions of existing HIPs, but also to lesser represented processes, including transcription and cell organization (Fig. EV2E,F). To explore the potential of HIPs to influence HD pathobiology, we investigated whether select targets could modulate mHTT-induced neuronal dysfunction phenotypes in a well-established *Drosophila* HD model (Onur et al, 2021). Fruit flies expressing an expanded N-terminal fragment of human HTT ($HTT^{NT231Q128}$) exhibit late-onset, progressive motor impairments, which can be quantified using the startle-induced negative geotaxis response (Onur et al, 2021). We evaluated whether manipulation of HIP genes affect longitudinal neuronal function in the animal (Fig. 8A). The genes assessed included the *Drosophila* orthologs of HIPs involved in transcription and chromatin remodeling (SMARCA5, CHD2, ELP3, MED15, MED27) and ion transport across membranes (KCNQ3/5, VDAC1/3, CACNA1, and CACNB) (Dataset EV5).

For some genes, such as the *Drosophila* ortholog of the L-type calcium channels (CACNB1/3), we reproduced previously reported genetic interactions that ameliorated mHTT-induced neuronal dysfunction using multiple alleles or constructs (Fig. 8B and Al-

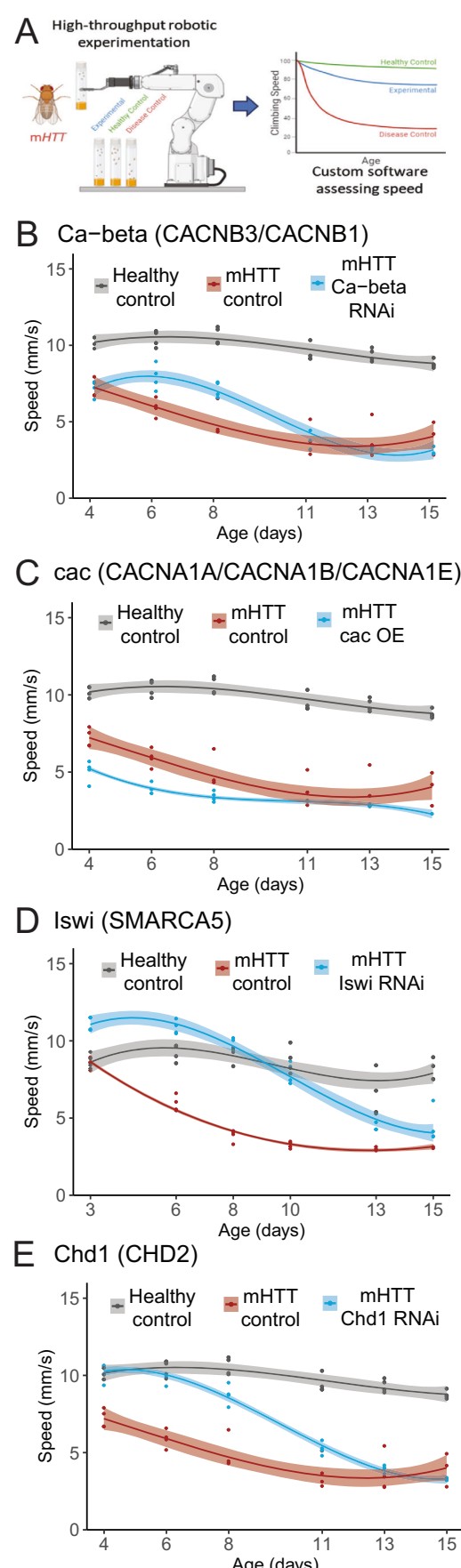

**Figure 8. Knockdown of HTT PPIs in an HD fly model identify mutant Huntingtin phenotype suppressors.**

(A) Schematic of fly motor performance assay. Assays identify genes that modify the progression of mutant Huntingtin neuronal dysfunction phenotypes. Representative graphical result of a fly motor performance assay performed for a target that is a genetic suppressor of mHTT phenotypes (experimental vs disease control) (B-E) Longitudinal plots of fly motor performance (speed in mm/s) as a function of age in wild-type flies (gray), mHTT-expressing flies (blue), and mHTT-expressing files with genetic manipulation of striatal-enriched HTT PPI (red) ($n = 4$ biological replicates). Differences between sample groups were determined by ANOVA (P value reported in Dataset EV5).

Ramahi et al, 2018). Broadly, several subunits of L-type calcium channels are known modifiers of HD phenotypes in humans and rodents (Fig. 7C). We found CACNA1A aggravated HD fly phenotypes when its ortholog was overexpressed (Fig. 8C). This aligns with previous research showing that partial loss of function of the *Drosophila* CACNA1A ortholog suppresses HD phenotypes (Al-Ramahi et al, 2018). Like HTT, CACNA1A is a CAG repeat-containing protein. In HD patients with below median pathological HTT CAG repeats, CACNA1A variants with larger CAG repeat lengths are linked to later age-of-onset (Stuitje et al, 2017). RNAi knockdown of VDACs, MED15, and MED27 resulted in lethality in the disease model, and the mediator subunit knockdowns were also lethal in healthy control flies (Dataset EV5). This is consistent with a previous study that found these are neuron-essential genes using a genome-wide mouse CNS genetic screen (Wertz et al, 2020). In contrast, knockdown of the *Drosophila* orthologs of the chromatin remodeling genes, SMARCA5 and CHD2, dependably suppressed HD fly phenotypes using multiple genetic alleles and constructs (Fig. 8D,E). Taken together, these in vivo genetic modifier assays strengthen the potential for these striatal-enriched, polyQ-dependent HIPs to be causal instigators of HD pathogenesis.

## Discussion

The knowledgebase of HTT interacting partners reflects over 3400 candidates (Aaronson et al, 2021). Association with these partners places huntingtin in several different cellular neighborhoods. Yet, this presents challenges in using this knowledge to understand its structure-function relationships, and how its interactions contribute to disease pathogenesis. Indeed, only a single interaction—Huntingtin-associated protein 40 (HAP40)—has been structurally characterized (Guo et al, 2018; Harding et al, 2021), with its disease relevance remaining under active investigation. Therefore, the arrangement and coordination of the myriad of HTT interacting partners remains poorly delineated. In this study, our use of antibodies targeting distinct epitopes within HTT allowed us to capture unique subcellular pools of HTT complexes. Proteomic characterization of these pools revealed a core set of interactions, as well as unique interactions, some of which showed a co-dependence between the targeted epitope and a specific tissue. For example, CREB-binding protein (CREBBP) was largely a consistent interaction, with the exception that it could not be isolated from the striatum using the N-terminal antibodies. Evaluating this striatum-selective, epitope-dependent phenotype more broadly, we found it was linked to interactions that had a functional enrichment in

nuclear annotations. Supporting this observation, our prior in-cell HTT PPI validation experiments using bioluminescence resonance energy transfer (BRET) often identified positive hits using only one of the N or C-terminal bait-prey combinations (Greco et al, 2022). This was suggestive of interactions that were constrained to specific domains/regions of HTT. With respect to the C-terminus, the occurrence of domain-specific interactions or long-distance interactions with the N-terminal domains (Vijayvargia et al, 2016) may occlude antibody epitopes and contribute to the poor performance of C-terminal HTT antibodies. Similarly, reduced co-isolation of nuclear-annotated interactions with antibodies targeting the N-terminus may be due to interactions that either directly bind to and/or occlude this domain. It is also possible that striatum-derived HTT-containing complexes have distinct structure–function relationships since reduced capture is usually not observed in the cortex or cerebellum. The broader issue of antibody preference for certain states/localizations of HTT is underappreciated. From this perspective, our study starts to define these epitope-dependent preferences and leverages the orthogonal results to have a more complete repertoire of HTT interacting partners. Although outside the scope of this study, targeted molecular and structural studies will be required to precisely localize the interaction interfaces, which may be facilitated by recent efforts from Alteen and colleagues to generate functional subdomains of HTT:HAP40 (Alteen et al, 2023).

Currently, an unmet need in the HD community is the shortage of molecular signatures that contribute to the vulnerability of striatal cells in HD. One piece of this puzzle is the growing consensus that cell type susceptibility to death is associated with its permissiveness to HTT CAG repeat expansion during disease progression (Lee et al, 2019; Monckton, 2021). In postmortem human samples from individuals with HD, the cells that are most sensitive to HD-induced death, striatal GABAergic medium-spiny projection neurons (MSNs) and cortical glutaminergic projection neurons, had the highest levels of expansion (Mätlik et al, 2024; Pressl et al, 2024). Interestingly, high levels of CAG expansion in the mutant ATXN3 gene have been observed in striatal MSNs from spinocerebellar ataxia 3 (SCA3) donors, but this did not result in cell loss (Mätlik et al, 2024). Therefore, somatic expansion alone is necessary but not sufficient for MSN cell death in HD. This suggests that the highest value HD molecular signatures may reflect striatal enriched, huntingtin-associated factors that synergize with somatic expansion processes to cause neuronal damage. For example, one hypothesis is that the highest value candidates are HTT interacting partners that are involved in DNA repair and maintenance pathways and are HD disease modifiers in humans. These stringent criteria draw focus to the delayed age of onset modifier, TCERG1, which was an interaction with wild-type and mutant HTT that was detected in the striatum but enriched in the cerebellum. Given that genetic variants of TCERG1 associated with HD are protective, it is tempting to speculate that its cerebellar interaction reflects this effect. Yet, the interaction of TCERG1 with mutant HTT in the striatum was detected at a low level, suggesting a more complex relationship between brain regions that warrants further investigation.

We also took an unbiased bioinformatics approach to assemble HD molecular signatures, drawing on new and existing knowledge of HTT interactions. One consideration is that a little over 1/3 of the HTT interaction knowledgebase is derived from various HD mouse models. While HD mouse models exhibit motor and cognitive impairments that mimic human disease, there are limitations. Caveats include differences in HTT gene sequences and structure, HD phenotypes, and disease progression of various mouse models (e.g., R6/2, YAC, and BACHD), as well as the inability to recapitulate more complex aspects of the human disease. Therefore, multiple lines of evidence are valuable when interpreting an interaction's disease relevance. Towards this goal, we focused on 165 striatum-enriched and polyQ-dependent interaction candidates. A promising aspect is that approximately a third of these have been reported as genetic HD modifiers in various HD models. The significant functional enrichment of these candidates in ion homeostasis and energy production, and potentiated intracellular calcium responses we observed with genetic deletion of HTT in neuroblastoma cells, suggest they are primary sensitizing pathways. Indeed, dysregulation of calcium signaling is well known in HD (Raymond, 2017; Czeredys, 2020), with the functions of store-operated calcium channels and their regulators affected by the presence of mHTT. Store-operated calcium entry (SOCE) repletes intracellular calcium stores. A primary regulator of SOCE is the ER membrane protein, STIM1, which senses ER $Ca^{2+}$ levels and subsequently targets Orai1 and TRPC1 calcium channels (Putney et al, 2017; Prakriya and Lewis, 2015). This process is aberrant in neuronal and mouse HD models, leading to elevated intracellular calcium levels (Wu et al, 2004; Tang et al, 2005). Related to SOCE, calcium-linked mitochondrial dysfunction has also been reported at the early stages of the disease prior to symptom onset (Milakovic et al, 2006; Panov et al, 2002), which is associated with direct binding of mHTT and impaired calcium handling (Milakovic et al, 2006; Panov et al, 2002). Moreover, expression of mutant HTT in mouse neuroblastoma and MSNs showed enhanced SOCE activation, which was reversed by RNAi knockdown of Orai1, TRPC1, or STIM1 (Vigont et al, 2015, 2014). In the present study, we identified the specific factors in calcium handling pathways that are disease-associated HTT interactions, including STIM1, which was a striatum-enriched, mHTT-dependent interaction. A previous study of HTT function in skeletal muscle suggested a link to calcium signaling via its interaction with STIM1 and/or Junctophilin 1 (JPH1), which points to involvement of the STIM1-ORAI1 complex and/or RyR1-DHPR coupling, respectively (Chivet et al, 2023). We did not detect an HTT:JPH1 interaction in the mouse brain, perhaps related to its preferential expression in skeletal muscle (Chivet et al, 2023). In addition, our identification of a mHTT-dependent interaction with the IP$_3$ receptor, ITPR1, points to IP$_3$-induced $Ca^{2+}$ from ER stores, which is upstream of SOCE activity. While STIM1-dependent SOCE activation is thought to support spontaneous neurotransmitter release, STIM1 has also been reported to influence dendritic spine structural plasticity through the inhibition of L-type $Ca^{2+}$ channels (Dittmer et al, 2017). This pathway may also be relevant in HD, as we observed several L-type channel subunits (CACNA1A-C, CACNB1-4) that were mHTT-dependent interactions, striatum enriched, and genetic modifiers in HD flies (Fig. 8). For example, the CACN beta subunits, which play a role in regulating the activity and assembly of the alpha subunits (Williams et al, 1992). Taken together, we postulate that STIM1 is a central player in disease pathogenesis, which could involve calcium signaling at rest (through IP$_3$R-dependent SOCE) and/or during activity-dependent synaptic plasticity (though $Ca^{2+}$-induced $Ca^{2+}$ release).

Our integration of experimental IP data with computational analysis of known CORUM complexes supports HTT's interaction with macromolecular complexes. One of the most prominent findings was co-isolation of HTT with nearly all 26 canonical Mediator complex subunits plus its associated kinase module. These subunits assemble the 1.2 MDa Mediator complex in mice (Youn et al, 2019; Dotson et al, 2000), which is known to engage with transcription factors and RNA Pol II for the initiation of transcription (Kim et al, 1994). For example, Mediator interacts with PPARg and SREBPs, which are primary regulators of lipid metabolism (Zhang et al, 2013). Impairments in cholesterol metabolism are well documented in HD, and at least one mechanism involves mHTT-dependent dysregulation of SREBP (Kacher et al, 2022). In this study, our data support the primary HTT:Mediator interaction is with Med15 and/or Med27 subunits within tail domain. As the tail domain interacts with SREBP, it is possible that dysregulation of SREBP in HD is caused by impaired HTT and Mediator association. The identification of Med15 is intriguing as it contains two polyQ repeat stretches (Zhu et al, 2015) and an overlapping prion-like domain that can mediate amyloid-like aggregation (Batlle et al, 2021; Zhu et al, 2015). The Med15 polyQ domain is likely functional as polyQ length can modulate its turnover rate and influence cellular responses to stress (Gallagher et al, 2020). Notably, a recent GWAS study identified a disease-hastening modifier at the MED15 locus in HD patients (Consortium et al, 2024). In our study, Mediator co-capture was highest in the striatum using antibodies targeting the proline-rich region or N-HEAT domain, and lowest with an N17-targeted antibody, suggesting the interaction is occurring within exon 1, possibly via mutual polyQ stretches. This is consistent with our observation that mutant HTT had altered interaction with Mediator and influenced the cellular dynamics of the Mediator complex, most strongly the Med15 subunit. Specifically, we observed a shift in Mediator protein abundance from the nucleus to the cytoplasm in the striatum of 40-week-old mice expressing mHTT. Additionally, we also found that the lack of HTT impacted the stability of several mediator subunits, most notably Med29, whose stability profile became more similar to Med15, a known interacting partner (Cho et al, 2022; Huttlin et al, 2021; Sato et al, 2004). Intriguingly, a previous study found that MED29 was the most disruptive single-subunit tail deletion, causing destabilization of the entire Mediator complex (El Khattabi et al, 2019). Taken together with our observation that HTT is largely cytoplasmic, we propose a model in which wild-type HTT is a positive regulator of Mediator's function in transcription pre-initiation, while mHTT disrupts this function by sequestering Med15 and likely other subunits in the cytoplasm, thus contributing to transcription-associated HD pathology. Our data pointing to Med15 as a mHTT effector in mice may be a conserved pathological mechanism in humans. The disease-hastening MED15 allele in HD individuals, rs177425, was shown to be in linkage disequilibrium with the longer, 13 CAG MED15 allele, suggesting the involvement of its polyQ repeat in the modifier effect (Consortium et al, 2024), which may facilitate a gain of toxic interaction with mutant HTT's expanded polyQ.

Functionally adjacent to transcription-linked complexes, the interaction of chromatin remodeling complexes was also influenced by mHTT. Yet in contrast to Mediator, the interaction of the ATP-dependent BAF (SWI/SNF) complex with mHTT was selectively reduced in the striatum (Fig. 3E). The BAF complex is linked to profound developmental neurological dysfunction and can be shuttled to the cytosol under cell-stress conditions (Alfert et al, 2019; Dastidar et al, 2012). It is tempting to speculate that the opposing effects of mHTT toxicity on its interactions with Mediator (transcription) and BAF (chromatin modifying) complexes confers functional significance. Indeed, recent efforts to understand the mechanistic links between transcription and chromatin accessibility suggest that BAF synergizes with RNA polymerase II to evict nucleosomes, with transcription factor binding mediating locus specificity (Brahma and Henikoff, 2024). While our study demonstrated mHTT- and localization-dependent alterations in HTT interaction with these complexes, it is challenging to determine if these effects are linked or occur in parallel without knowing the downstream impact on the complex activities. We explored the disease relevance of the mediator interaction using genetic HD modifier assays, but knockdown of Drosophila homologs of MED15 and MED27 were lethal in HD flies, likely due to their essentiality (Wertz et al, 2020). From the perspective of other chromatin remodeling factors, knockdown of either SMARC5 or CHD2 ameliorated HD phenotypes. Yeast homologs of SMARCA5 and CHD2 interact with each other (Gavin et al, 2002) and, in humans, SMARCA5 is a component of multiple chromatin remodeling complexes, including ISWI, which has roles in brain development (Goodwin and Picketts, 2018) and DNA repair (Aydin et al, 2014) and B-WICH/WICH, which regulates histone H3K9 acetylation by recruiting lysine acetyltransferases to DNA (Vintermist et al, 2011). CHD2 is a helicase that binds target gene promoters and may facilitate histone 3.3 deposition. Given the broad cellular roles of these complexes, future studies will be required to understand if and how the coordination of nucleosome remodeling activities, RNAPII-mediated transcription, and DNA damage/repair contribute to HD pathogenesis.

# Methods

**Reagents and tools table**

| Reagent/resource | Reference or source | Identifier or catalog number |
|---|---|---|
| **Experimental models** | | |
| B6J.129S1-Htt^tm1Mfc/140ChdiJ (Mus musculus) | Jackson Lab | RRID: IMSR_JAX:027409 |
| B6.129S1(Cg)-Htt^tm2Mem/20ChdiJ | Jackson Lab | RRID: IMSR_JAX:027411 |
| Human BE(2)-C cells | ATCC | CRL-2268 |
| Drosophila mutant HTT strains | This study | Dataset EV5 |
| **Antibodies** | | |
| Mouse anti-HTT (2B7) | Coriell Institute | RRID:AB_3096092 |
| Mouse anti-HTT (4C9) | Coriell Institute | CH03157 |
| Rabbit anti-HTT (138) | Coriell Institute | CH00147 |
| Rabbit anti-HTT (139) | Coriell Institute | CH00168 |
| Mouse anti-HTT (4E10) | Santa Cruz Biotechnology | RRID:AB_627768 |
| Mouse anti-HTT (3E10) | Santa Cruz Biotechnology | RRID:AB_627767 |
| Mouse anti-HTT (8A4) | Santa Cruz Biotechnology | RRID:AB_627769 |
| Rabbit anti-HTT (EPR5526) | Abcam | RRID:AB_10863082 |
| Rabbit anti-HTT (D7F7) | Cell Signaling Technology | RRID:AB_10827977 |

| Reagent/resource | Reference or source | Identifier or catalog number |
|---|---|---|
| Rabbit anti-HTT (D7F7) | Cell Signaling Technology | #31873 |
| **Oligonucleotides and other sequence-based reagents** | | |
| sgRNA HTT Exon 1 5'-TTGTCAGACAATGATTCACA-3' | Invitrogen | A35533 |
| sgRNA HTT Exon 2: 5'-GAAGGACTTGAGGGACTCGA | Invitrogen | A35533 |
| sgRNA HTT Exon 3: 5'-CCCAGAAGTTTCTGAAATTC-3 | Invitrogen | A35533 |
| scrambled negative-control sgRNA | Invitrogen | A35526 |
| CRISPRMAX transfection reagent | ThermoFisher Scientific | CMAX0001 |
| TrueCut Cas9 Protein V2 | Invitrogen | A36499 |
| **Chemicals, enzymes, and other reagents** | | |
| Protein A/G magnetic beads | ThermoFisher Scientific | 88802 |
| HALT protease and phosphatase inhibitor | ThermoFisher Scientific | 78446 |
| Universal nuclease | ThermoFisher Scientific | 88700 |
| BCA assay | ThermoFisher Scientific | 23225 |
| NE-PER reagents | ThermoFisher Scientific | 78833 |
| Bond Breaker TCEP | ThermoFisher Scientific | 77720 |
| 2-Chloroacetamide | Millipore Sigma | C0267 |
| n-dodecyl-β-maltoside | ThermoFisher Scientific | 89902 |
| Triethylammonium bicarbonate buffer | Millipore Sigma | T7408 |
| Trypsin protease, MS Grade | ThermoFisher Scientific | 90059 |
| SDB-RPS StageTips | CDS Analytical | 98-0604-0226-4EA |
| HeLa Protein Digest Standard | Thermo Fisher Scientific | 88329 |
| Formic Acid | Thermo Fisher Scientific | 28905 |
| Methanol LC/MS grade | Thermo Fisher Scientific | A456-4 |
| Acetonitrile LC/MS grade | Thermo Fisher Scientific | A955-4 |
| Water UHPLC-MS grade | Thermo Fisher Scientific | W81 |
| goat anti-mouse IgG AlexaFluor488 | Thermo Fisher Scientific | A-11029 |
| goat anti-rabbit IgG AlexaFluor568 | Thermo Fisher Scientific | A-11036 |
| **Software** | | |
| MSConvert, v3.0 | https://proteowizard.sourceforge.io/ (Chambers et al, 2012) | |
| FragPipe, v18.0 & v20.0 | https://fragpipe.nesvilab.org (Kong et al, 2017) | |
| REPRINT / SAINTexpress | https://reprint-apms.org/ (Teo et al, 2014) | |
| DIA-NN v1.8.1 | https://github.com/vdemichev/DiaNN (Demichev et al, 2019) | |
| Skyline v21.2 | https://skyline.ms/project/home/software/Skyline/begin.view (MacLean et al, 2010) | |
| ClustVis | https://biit.cs.ut.ee/clustvis/ (Metsalu et al, 2015) | |
| Tapioca | https://github.com/FunctionLab/tapioca (Reed et al, 2024) | |
| Fiji (Image J | https://imagej.net/software/fiji/downloads (Schindelin et al, 2012) | |

| Reagent/resource | Reference or source | Identifier or catalog number |
|---|---|---|
| **Other** | | |
| High pH Reversed-Phase Peptide Fractionation Kit | Thermo Fisher Scientific | 84868 |
| S-Trap micro filter digestion devices | Protifi, Inc | C02-micro-80 |
| Q-Exactive HF | ThermoFisher Scientific | N/A |
| Ultimate 3000 nRSLC | ThermoFisher Scientific | N/A |
| EASYSpray C18 column | ThermoFisher Scientific | ES903 |
| timsTOF Ultra | Bruker Daltonics | N/A |
| nanoElute 2 | Bruker Daltonics | N/A |
| PepSep ULTRA column | Bruker Daltonics | 1893484 |
| Nikon Ti-E confocal microscope/Yokogawa spinning disc | Nikon Ti-E/CSU-21 | N/A |
| Synergy H1 microplate reader | BioTek | N/A |
| FURA-2-AM assay | Thermo Fisher Scientific | F1221 |

## Mouse models

Mice were housed by Jackson's labs, which ensure the ethical and humane treatment of animals.

Cortex, cerebellum, and striatal tissues were dissected by Jackson Laboratories from 8 and 40-week-old male or female homozygous chimeric HTT: human HTT exon 1 (Hdh:HD exon 1) mice encoding the human version of the polyglutamine/polyproline-rich segment with either 18 CAG repeats (strain #027411) or ~140 pure CAG repeats (strain #027409).

## Antibodies

The following anti-HTT antibodies were used: 2B7 (CH03023), 4C9 (CH03157), 138 (CH00147), and 139 (CH00168) from the Coriell Institute; 4E10 (sc-47758), 3E10 (sc-47757), and 8A4 (sc-47759) from Santa Cruz Biotechnology; EPR5526 (ab109115) from Abcam; and D7F7 (#5656 and #31873) from Cell Signaling Technology. Non-specific IgG antibodies were used as isotype controls to match the host species of the target antibody.

## Magnetic bead conjugation

Antibodies were pre-conjugated to magnetic protein A/G beads (ThermoFisher Scientific, 88802) for 1 h at 4 °C in 20 mM K-HEPES pH 7.4, 110 mM KOAc, 2 mM MgCl$_2$, 1 μM ZnCl$_2$, 1 μM CaCl$_2$, 150 mM NaCl, and 0.1% Tween-20. For full IP-MS experiments, we utilized 25 μg of antibody per IP that was conjugated to a 25 μl slurry of beads per 2 mg of input supernatant. Following an overnight incubation at 4 °C, beads were washed three times in 500 μL IP Buffer [20 mM K-HEPES pH 7.4, 110 mM KOAc, 2 mM MgCl$_2$, 1 μM ZnCl$_2$, 1 μM CaCl$_2$, 150mM NaCl, 0.1% Tween-20, and 1% Triton X-100], and suspended in 50 μL IP buffer. This buffer composition was selected based on its extensive prior assessment and implementation. It was designed to minimize disruption of cellular complexes and provide ion cofactors to maintain various enzymatic activities (Cristea et al, 2005; Joshi et al, 2013; Mathias et al, 2016), and with some modifications, it was

previously optimized for the capture of FLAG-HTT complexes (Greco et al, 2022). The non-denaturing Triton X-100 detergent provides reasonable protein extraction while maintaining interactions within lipid rafts (Kramer et al, 2012; Miteva et al, 2013), which is relevant given HTT can be membrane-associated (Kegel et al, 2005).

## Tissue lysis

For each biological replicate, mice of the same sex were pooled (for striatum) on a per replicate basis with replicates 1 and 3 being composed on female mice and replicates 2 and 4 being composed of male mice. The tissue was minced on ice and then lysed in 4 mL of IP buffer (see above), supplemented with 1:100 dilution of HALT protease and phosphatase inhibitor (Thermo Fisher Scientific, 78446) and 1:2500 dilution of universal nuclease (Thermo Fisher Scientific, 88700) by homogenizing in a Tenbroeck homogenizer for 20 strokes on ice, then incubation on ice for 5 min. The lysates were cleared at $8000 \times g$ for 10 min at 4 °C to separate soluble protein (supernatant) from insoluble material (pellet). The resulting supernatant yielded on average ~9 mg of protein as determined by BCA assay (Thermo Fisher Scientific, 23225). The lysates were aliquoted (2 mg per IP) and diluted to 2 mg/mL.

## Immunoaffinity purification (IP) of HTT

Antibody-conjugated beads were combined with lysates (2 mg) and incubated for 1 h at 4 °C while rotating. Following the 1-h incubation, the beads were collected using a magnet, the resulting flow through was saved, and then the beads were resuspended in 1 ml of IP wash buffer (IP buffer without protease and phosphatase inhibitors) and transferred to a 1.5-ml round bottom tube. The beads were then washed $3 \times 1$ ml IP wash buffer, resuspended in 1 ml of cold ddH$_2$O and transferred to a new 1.5-ml tube. Beads were washed once more with 1 ml of cold ddH$_2$O and then eluted in 50 µL TES buffer (106 mM Tris HCl, 141 mM Tris base, 5% SDS, 0.5 mM EDTA) at 70 °C for 10 min. Beads were magnetically separated from the eluted proteins (supernatant), and the eluted proteins were then transferred to a new 1.5-ml tube.

## Preparation of IP samples for liquid chromatography-tandem mass spectrometry (LC-MS/MS)

Eluted proteins (50 µL) and input proteins (50 µg) were reduced and alkylated by TCEP and CAM at final concentrations of 25 mM and 50 mM, respectively, at 70 °C for 20 min. The proteins were digested using S-Trap spin columns (Protifi). Briefly, samples were adjusted to 1.2% phosphoric acid using a 12% phosphoric acid solution. S-trap binding buffer (165 µL of 100 mM TEAB, pH 7.1, 90% MeOH) was added to each sample, applied to S-trap column, centrifuged briefly at 4000×g, washed 2× 150 µL S-trap binding buffer, 2× 100 µL methanol:chloroform:water (4:1:4), and 5 × 150 µL S-trap binding buffer. Proteins were digested with trypsin (1 µg) in 20 µL of 50 mM TEAB, pH 8.0, for 1 h at 42 °C with no shaking. The resulting peptides were eluted sequentially with 50 µL of 50 mM TEAB, pH 8.0, 50 µL of 0.2% FA, and 50 µL 0.2% FA in 50% ACN, with centrifugation at 4000×g between elutions. Samples were dried down by vacuum-assisted centrifugation, then resuspended in 5 µL of 1% FA/1% ACN.

## LC-MS/MS analysis of IP and proteome samples

IP and total proteome (input) samples digested above were analyzed on a Q-Exactive HF mass spectrometer equipped with an EASYSpray ion source coupled directly to an Ultimate 3000 nRSLC HPLC (Thermo-Fisher Scientific). Peptides (~1 µg) were separated using reverse-phase chromatography over an EASYSpray C18 column (2 µm particle size, 75 µm diameter, 500 mm length, heated to 50 °C) (ThermoFisher Scientific, ES903) for a 120 min (IP) or 150 min (input) linear gradient from 3 to 25% Solvent B (0.1% formic acid in 97% LC-MS grade acetonitrile/2.9% LC-MS grade water) at a flow rate of 250 nL/min. MS and MS/MS spectra were acquired in data-dependent (IP) or data-independent (input) acquisition mode. Data-dependent acquisition (DDA) was performed as follows. A full MS1 scan (350–1800 $m/z$) was collected at a resolution of 120,000, with a target AGC of 3e6 and maximum injection time of 30 ms, followed by data-dependent MS2 scans on the top 20 most intense precursor ions using higher energy collision-induced dissociation (HCD), with a 45 s dynamic exclusion duration. MS2 isolation windows were set to 1.2 $m/z$, with a target AGC of 1e5 and a maximum inject time of 42 ms, a resolution of 15,000, and normalized collision energy (NCE) of 28. Data-independent acquisition (DIA) was performed using instrument settings adapted from Pino et al, 2020. Briefly, MS1 scans were acquired in centroid mode at a resolution of 60,000, an AGC target of 3e6, a maximum IT of 55 ms, a scan range of 395–1005 $m/z$. DIA MS2 scans were acquired in an overlapping/staggered DIA window scheme, which consisted of $25 \times 24$ m/z wide isolation windows covering 400–1000 $m/z$, collected in centroid mode at a resolution of 30,000, an AGC of 3e6, a maximum IT of 55 ms, a loop count of 25, an isolation window of 24.0 $m/z$, and an NCE of 29.

## Bioinformatics analysis of IP and proteome data

Instrument raw data from IP experiments were converted to mzml using MSConvert (Proteowizard ver. 3.0). Mass and retention time recalibration, peptide-spectrum scoring (MSFragger and Percolator), peptide-to-protein summarization (Philosopher), and label-free protein quantification (IonQuant) were performed using the default LFQ workflow in FragPipe (ver. 18.0), except MaxLFQ was disabled and peptide match-between-run FDR was set at 2%. Briefly, experimental spectra were searched against a mouse sequence database (UniProt-reviewed, downloaded 2022-04 in FragPipe) appended with common contaminants (20,278 total unique sequences), which were reversed and concatenated for the MSFragger database search. Database searching was performed default settings, including fixed modification of cysteine carbamidomethylation, and variable modifications of methionine oxidation and N-terminal protein acetylation. Specific HTT interaction candidates were assigned by the SAINTexpress algorithm (Teo et al, 2014) using the default settings implemented on the Reprint-APMS server (https://reprint-apms.org/). In preparation for SAINTexpress analysis, the Reprint spectral count (SPC) output files generated by FragPipe were pre-processed: (1) contaminants entries removed, (2) for each protein in the bait IPs, the replicates with the two highest spectral counts within sample groups were selected, and (3) a virtual negative control set was generated for each protein in the IgG controls ($n = 37$ replicates across sample groups and tissues) by selecting the values reflecting the middle 10 highest SPC. Following SAINTexpress, the top candidates HTT

interactions were retained by filtering for BFDR ≤1% and an average bait SPC ≥5. Comparisons of bait and prey abundances were performed using raw LFQ values calculated by FragPipe, while correlation and PCA analyses were performed using FragPipe-normalized LFQ intensities. Mzml files from total proteome DIA experiments were processed by DIA-NN (Demichev et al, 2019) by searching against a predicted spectral library generated in silico from a mouse sequence database (UniProt-reviewed, download 2023-03) appended with common contaminants (17,272 total unique sequences). The predicted spectral library was generated using the default DIA-NN settings, specifying peptide length of 7–30 amino acids, precursor charge state of +1 to +4, a precursor *m/z* range of 300–1100, and a fragment ion *m/z* range of 200–1800. Cysteine carbamidomethylation was specified as a fixed modification and methionine oxidation and protein N-terminal acetylation were set as variable modifications. RT-dependent cross-run normalization was performed with MBR enabled with mass accuracy, MS1 accuracy, and scan window settings specified as 12 ppm, 8 ppm, and 12, respectively. The default DIA-NN protein group matrix, which is filtered to 1% protein FDR, was used for downstream analysis.

## Targeted PRM-MS/MS validation of PPIs

Two replicates from IP-MS experiments were analyzed by Parallel Reaction Monitoring (PRM)-MS/MS. Peptides were separated and analyzed as in "*LC-MS/MS Analysis of IP and Proteome Samples*", described above, except peptide separation was performed over 60 min, and an inclusion list was used to sequentially target peptides for MS/MS fragmentation. To assist in peak peaking, a spectral library was built in Skyline (MacLean et al, 2010) using the PRM data searched in FragPipe (interact.pep.xml files). Thermo-Raw PRM files were imported into Skyline to extract fragment ion chromatograms (PRM setting @ 60,000 resolution). Peak picking, transitions for quantification, and integration boundaries were manually verified. The Skyline analysis document and supporting files were uploaded to PanoramaWeb.

## Preparation and LC-MS/MS analysis of nuclear and cytosolic fractions from striatum

Nuclear-cytoplasmic fractionation was performed in triplicate using the NE-PER kit (Thermo Fisher Scientific), as per the manufacturer's instructions, on frozen mouse striatum for the 4 genotype-age samples analyzed in the IP-MS experiments (Q20-8 weeks, Q140-8 weeks, Q20-40 weeks, 140-40 weeks). Following biochemical fractionation, the protein yield was determined by the BCA assay. In total, 50 µg of protein was used for trypsin digestion, as described above in "Preparation of IP Samples for LC-MS/MS". Samples were generated for spectral library generation by separately pooling aliquots of the cytoplasm and nuclear-enriched fractions, which were then separated into eight analytical fractions by basic reverse-phase spin columns (Thermo Fisher Scientific). LC-MS/MS analysis (1 µL of sample containing 150 ng of peptides on column) was performed on a nanoElute2 coupled to a timsTOF Ultra mass spectrometer (Bruker). The mobile phases were 0.1% LC-MS grade FA (Thermo Fisher Scientific) in 99.9% UHPLC-MS water (Fisher Scientific, buffer A) and 0.1% FA in 99.9% UHPLC-MS acetonitrile (Fisher Scientific, buffer B). A one column method was used with a

PepSep ULTRA (25 cm × 75 µm × 1.5 µm) C18 HPLC column (Bruker) and a 10 µm emitter (Bruker) attached to a CaptiveSpray Ultra source with a column toaster set to 50 °C. For analysis of pooled samples for spectral library generation, a linear 90-min gradient from 4% to 35% buffer B at a flow rate of 150 nL/min was used for peptide separation, over which a data-dependent acquisition-parallel accumulation serial fragmentation (dda-PASEF) method was employed for peptide analysis. The MS1 scan settings were set at 100–1700 *m/z* in positive ion polarity. For the TIMS settings, the mode was set to custom, with a start and end $1/K_0$ of 0.65 V·s/cm$^2$ and 1.46 V·s/cm$^2$, respectively. The ramp time was set to 50 ms with a 100% duty cycle and a ramp rate of 17.80 Hz. For analysis of individual biological replicates, a linear 30-min gradient from 3 to 34% buffer B at a flow rate of 200 nL/min was used for peptide separation, over which a dia-PASEF method was employed for peptide analysis. The detector settings were the same as above, except variable width DIA windows (3 × 16 design) were specified based on optimization using the py_diAID tool (Skowronek et al, 2022) on dda-pasef analysis of HeLa peptide standards (Thermo Fisher Scientific). The optimized windows are provided in Table EV1.

## BE(2)-C cell culture and CRISPR knockout

Human BE(2)-C cells (ATCC # CRL-2268) were cultured in F-12 medium mixed 1:1 with MEM and supplemented with 10% FBS. Mycoplasma screening was performed using mass spectrometry-based detection. Samples analyzed by LC-MS/MS analysis (see below) were routinely searched against a human protein sequence database that was appended with the UniRef50 sequences for small ribosomal subunit protein uS9 (UniRef50_Q98Q72), DNA-directed RNA polymerase subunit beta (UniRef50_Q98Q23), and isoleucine–tRNA ligase (UniRef50_Q98PQ2). These entries represent proteins with at least 50% sequence homology that are shared among the greatest number of Mycoplasmopsis family members (487–635 members), including those that most commonly infect cell cultures. HTT knockout was performed as previously described (Justice et al, 2021) using the TrueCut system (ThermoFisher). Briefly, cells were seeded the evening before transfection at $0.89 × 10^5$ cells/ml. Three total HTT sequence-specific TrueGuide single-guide RNA (sgRNA) (Invitrogen #A35533) targeting Exon 1: 5'-TTGTCAGACAATGATTCACA-3'; Exon 2: 5'-GAAGGACTT-GAGGGACTCGA-3'; and Exon 3: 5'-CCCAGAAGTTTCT-GAAATTC-3; and scrambled negative-control sgRNA (Invitrogen #A35526) were combined in equal molar ratios with TrueCut Cas9 Protein V2 (Invitrogen #A36499) in Opti-MEM (Thermo Fisher Scientific #31985062) and 2:1 CRISPRMAX transfection reagent (Thermo Fisher Scientific #CMAX0001). Cells were allowed to recover, and KO efficiencies were determined by data-dependent label-free quantification mass spectrometry analysis, as described above for the analysis and processing of IP samples.

## Mediator complex dynamics in BE(2)-C cells using thermal proximity coaggregation (TPCA) assays

BE(2)-C cells (HTT SCR or HTT KD) were cultured as above in triplicate and were subjected to 5 temperature thermal denaturation (Reed et al, 2024). Cells were grown to 70% confluence in 15-cm plates, harvested with trypsin for 5 min, washed with media, pelleted at 300

rpm for 5 min, washed with 1X PBS, and pelleted again at 300 rpm for 5 min. Cells were then resuspended in 1X PBS and aliquoted into six PCR tubes, five of which would be subjected to thermal denaturation (37 °C, 40.7 °C, 44.6 °C, 52.8 °C, and 55.3 °C), while the remaining sample was lysed in 5% SDS as a reference proteome. After samples were subjected to thermal denaturation for 5 min, an equal volume of 1.5× kinase buffer (75 mM K-HEPES, pH 7.5, 15 mM MgCl$_2$, 1× HALT protease inhibitor cocktail, 3 mM TCEP) was added. Samples were then subjected to lysis with IP-DOC Buffer (20 mM K-HEPES, pH 7.4, 110 mM potassium acetate, 2 mM MgCl$_2$, 0.1% Tween-20, 1 µM ZnCl$_2$, 1 µM CaCl$_2$, 1% Trition-X-100, 0.5% sodium deoxycholate, 250 mM NaCl) by incubating on ice for 1 h with vortexing every 10 min. Thermally denatured proteins were pelleted from the lysates by centrifugation at 20,000×$g$ for 20 min at 4 °C, and the supernatant was transferred to a new tube. Reference proteome samples were lysed in 5% SDS and sonicated three times with 10 pulses and heated for 5 min at 95 °C. Samples were then reduced and alkylated with 25 mM TCEP and 50 mM CAM, frespectively, and subjected to methanol/chloroform precipitation. Protein pellets were resuspended in 160 µL of 100 mM K-HEPES, pH 8.3, and protein estimation was performed with the BCA assay (Thermo Fisher Scientific) on the reference and 37 °C samples. From the protein measurements, 50 µg of protein was aliquoted, while for the remaining temperatures, an average volume equivalent was removed from each of the remaining temperatures to preserve the effect temperature-dependent denaturation. All samples were adjusted to 100 µL and digested with trypsin at an estimated ratio of 1:50 trypsin:protein for 16 h at 37 °C with shaking at 600 rpm. Peptides were desalted with SDB-RPS (Fisher Scientific), and peptide concentrations were estimated by the Scopes method using a Nanodrop device. 7.5 µg of peptides were aliquoted from the reference and 37 °C samples into an autosampler vial, and an equal average volume equivalent were aliquoted into vials for the remaining temperatures. The reference and 37 °C samples were adjusted to 150 ng/µL with 0.1% FA in 4% ACN supplemented with 0.015% n-dodecyl-β-maltoside (Thermo Fisher Scientific) and remaining temperature sample volumes adjusted as described above.

## LC-MS/MS analysis of thermal proximity coaggregation experiments

LC-MS/MS analysis (1 µL of sample containing 150 ng of peptides on the column for reference and 37 °C; less for higher temperatures) was performed on a nanoElute2 HPLC coupled to a timsTOF Ultra mass spectrometer (Bruker). HPLC and dia-PASEF acquisition parameters were the same as described above in "Preparation and LC-MS/MS analysis of nuclear and cytosolic fractions from striatum".

## Bioinformatic processing of Bruker timsTOF DIA data

Bruker.d DIA data files were directly analyzed by DIA-NN (Demichev et al, 2019) using either an experiment-specific spectral library (for nuclear-cytoplasmic fractions from mouse striatum) or an in silico human spectral library (for TPCA assays). For the experiment-specific library, FragPipe (ver. 20) was used for peptide and protein identification (MSFragger v3.8/Philosopher 5.0) and spectral library generation (EasyPQP) using the search settings described above in "Bioinformatics analysis of IP and proteome data". For DIA-NN analysis of TPCA data, the human library in ProteoScape was used (Bruker, sequence database downloaded on

02/05/2024 and generated 02/14/2024). Cross-run normalization was disabled, while MBR was enabled. Mass accuracy and MS1 accuracy were fixed at 20 ppm and 15 ppm, respectively. The report.pg_matrix, report.pr_matrix, and the report.stats output files were utilized for downstream analysis with a modified version of Tapioca, a machine learning framework for TPCA data analysis (Reed et al, 2024).

## Immunofluorescence microscopy

BE(2)-C cells were seeded onto autoclaved 15-mm glass coverslips (VWR) at $1.5 \times 10^5$ cells/mL. The next day, cells were fixed in 4% PFA for 15 min at RT and washed three times with 1× PBS with 0.2% Tween-20 (PBST). Cells were permeabilized in 0.1% Triton X-100 in PBST for 15 min at RT, washed twice with PBST and then blocked in 2.5% human serum and 2% BSA in PBST for 60 min. Samples were incubated for 1 h with the primary antibodies used for the immunopurification studies diluted into blocking buffer. After primary staining, cells were washed 3× in PBST then incubated with DAPI (1:1000, Thermo Fisher Scientific) and appropriate secondary antibody diluted 1:2000 in block for 1 h: goat anti-Ms IgG highly cross-adsorbed AlexaFluor488 (1:2000, Thermo Fisher Scientific) or goat anti-Rb IgG cross-adsorbed AlexaFluor568 (1:2000, Thermo Fisher Scientific). Coverslips were mounted using Prolong Diamond and were imaged in the Princeton Confocal Imaging Core using an inverted fluorescence confocal microscope (Nikon Ti-E) equipped with a Yokogawa spinning disc (CSU-21) and digital CMOS camera (Hamamatsu ORCA-Flash TuCam) using a Nikon ×60 Plan Apo objective (maximum projections centered around the nuclear center). Image analysis was performed using FIJI (Image J).

## Intracellular calcium measurement

Relative intracellular calcium levels were determined by the FURA-2-AM assay (ThermoFisher #F1221) according to the manufacturer's protocol. Briefly, a 5mM stock was prepared in DMSO prior to use. BE(2)-C cells were cultured as above in a 96-well plate (2 x10$^4$ cells/well) overnight prior to labeling. Adherent cells were washed twice with PBS, then incubated with 5µM FURA-2-AM in PBS for 60 min. Cells were washed twice with PBS and then incubated for a further 30 min to allow de-esterification of the dye. Relative quantification of calcium levels was determined by ratiometric quantification of fluorescent intensity following excitation at 340 nm and 380 nm and detection at 510 nm on a Synergy H1 microplate reader (BioTek). After baseline measurements, cells were incubated with 1 µM FCCP to trigger calcium flux from mitochondrial stores (Jadiya et al, 2019). Intracellular calcium levels were measured again at 3 min post FCCP exposure.

## Drosophila strains and motor performance assay

*Drosophila* experiments were carried out using strains expressing mutant HTT (*UAS-HTT^{NT231Q128}*) under control of the panneural *elav*-GAL4 driver obtained through the Bloomington Drosophila Stock Center. *Drosophila* orthologs of candidate mice genes were identified using DRSC Integrative Ortholog Prediction Tool (Hu et al, 2011), filtering for the best match ortholog candidate. Mutant alleles and RNAi strains were obtained from both BDSC and the Vienna Drosophila Resource Center (VDRC). All experimental genotypes used are summarized in Dataset EV5. To assess neuronal dysfunction, we used an automated behavioral assay that takes advantage of the inherent startle-induced negative geotaxis response in *Drosophila*. As

previously described (Onur et al, 2021), we assessed motor performance of healthy control (*elav-GAL4; UAS-non-targeting*[hpRNA]), disease model (*elav-GAL4; UAS-HTT*[NT231Q128]*/UAS-non-targeting*[hpRNA]), and experimental group flies expressing mutant HTT and the allele or construct of interest as a function of age. We used ten age-matched virgin females across four replicates per genotype. Flies were collected within a 24-h period post-eclosion and transferred to new vials containing fresh media every other day. Using an automated robotic platform, the flies were taped to the bottom of a plastic vial and video-recorded for 7.5 s. The videos were analyzed using custom software to assess the speed of each individual fly. Three trials per replicate were performed each day for approximately two weeks. A non-linear mixed-effect model ANOVA, followed by a post-hoc test for pairwise comparisons with Holm correction, was used to assess statistical significance across genotypes.

## Experimental design

IP-MS experiments had three to five biological replicates per sample group. Exclusion of single replicates within the sample groups was considered by evaluating their Pearson correlation and variance by principal component analysis. Proteome and sub-proteome analyses were performed in three to four biological replicates per sample group, and their LC-MS/MS analysis was randomized. No sample blinding was used.

# Data availability

The datasets produced from this study are available in the following databases: Proteomics data: MassIVE MSV000094509 (https://massive.ucsd.edu/ProteoSAFe/dataset.jsp?task=2a89f97ea8a24bbb93bee9c56f9ea019). Targeted mass spectrometry data: Panorama-Web PXD051407 (https://panoramaweb.org/eHTT_PPI_PRM.url).

The source data of this paper are collected in the following database record: biostudies:S-SCDT-10_1038-S44320-025-00096-3.

# Peer review information

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

## Acknowledgements

The authors would like to thank Thomas Vogt and Brinda Prasad for their fruitful discussions about experimental design and data interpretation. IMC is grateful for funding from the CHDI Foundation, USA (Award A-18331) and TJR for support from the NSF GRFP (DGE-2039656). JB is supported by AG/NIA/NIH R01 AG057339.

## Author contributions

**Joshua L Justice**: Conceptualization; Data curation; Formal analysis; Validation; Investigation; Visualization; Methodology; Writing—original draft; Writing—review and editing. **Todd M Greco**: Conceptualization; Data curation; Formal analysis; Funding acquisition; Validation; Investigation; Visualization; Methodology; Writing—original draft; Writing—review and editing. **Josiah E Hutton**: Formal analysis; Investigation; Methodology; Writing—original draft. **Tavis J Reed**: Formal analysis; Investigation; Methodology. **Megan L Mair**: Formal analysis; Investigation; Visualization; Methodology; Writing—original draft. **Juan Botas**: Conceptualization; Resources; Formal analysis; Supervision; Funding acquisition. **Ileana M Cristea**: Conceptualization; Formal analysis; Supervision; Funding acquisition; Writing—original draft; Project administration; Writing—review and editing.

Source data underlying figure panels in this paper may have individual authorship assigned. Where available, figure panel/source data authorship is listed in the following database record: biostudies:S-SCDT-10_1038-S44320-025-00096-3.

## Disclosure and competing interests statement

The authors declare no competing interests.

# Expanded View Figures

**Figure EV1. Evaluating IP efficiency and specificity of HTT antibodies for isolation of endogenous WT and mutant HTT from whole brain lysates of HD mouse models.** ▶

(**A**) Domain map of HTT with IP antibodies marking their approximate epitope regions (see Materials and Methods). Blue cylinders represent the 7 HEAT repeat domains of HTT. Antibody IP efficiency was evaluated in Q20-8 weeks whole brain by western blotting using Odyssey Infrared detection (*bottom*). Antibodies in blue text had sufficient IP efficiency for further evaluation. FT = flow-through fraction, E = eluted fractions. (**B**) Western blot densitometry quantification of percent of HTT depleted from the flow-through (FT) vs. mIgG control lanes, and amount of HTT recovered from the elution (E) vs. mIgG control lanes. Antibodies are ordered (top to bottom) by epitope location (N-terminal to C-terminal). (**C**) Heatmap of the top 5% most abundant HTT interacting proteins ($n = 79$) isolated and quantified by IP-MS from whole brain using different HTT antibodies (EPR5526, 2B7-4C9, 4E10-3E10, 138-129). Raw MS protein abundances were normalized by their respective replicate medians and theoretical number of tryptic peptides and expressed as the $\log_2$ average of two biological replicates in Q20 and Q140-8 weeks conditions. Normalized protein abundances were analyzed by hierarchical clustering (distance metric: Euclidean and method: Ward) with no additional normalization or scaling. Missing values were imputed with the minimum quantified value. (**D**) Violin plot of $\log_2$ enrichment ratio (HTT IP vs mIgG IP) distribution for proteins co-isolated with HTT Q20 or Q140 from the striatum (8 weeks) using antibodies EPR5526, 2B7-4C9 pair, 4E10-3E10 pair, and D7F7. Enrichment ratio of HTT bait is indicated by orange circle. (**E**) Comparison of Htt, Hap40 (F8a1), and Synj1 levels in IPs of HTT Q20 from the striatum of 8 weeks old mice using EPR5526, 2B7-4C9, or 4E10-3E10 antibodies ($n = 4$, mean ± SD). Protein abundances were calculated as in (**C**). (**F**) Sequence alignment of Htt N17, corresponding to the EPR5526 epitope region, and Synj1 (aa. 399–408). The predicted crystal structure (AlphaFold 2.0) is shown, labeled with selected aligned amino acids of Synj1/HTT in an alpha-helix (dark blue), which forms hydrogen bond contacts with a proximal alpha-helix (light blue). (**G**) Scatterplot comparisons of co-isolated (prey) protein $\log_2$ enrichment ratios (HTT vs mIgG IP) in Q20 vs Q140 for N-terminal (left) and Central (right) antibody sets. Bait HTT and selected positive control interactions are denoted in red and orange circles, respectively. HTT interactions that were annotated in at least 2/3 of the previously reported HTT IP-MS studies from mouse brain are indicated in light blue circles. (**H**) Scatterplot of $\log_{10}$ abundance (MS summed intensity) for HTT vs. HAP40 showing high correlation in all individual IP-MS experiments by tissue. (**I**) Principal component analysis of all co-isolated $\log_2$ protein abundances stratified by antibody set (N-terminal and Central) and tissue. Abundances were median normalized. (**J**) Same as (**I**), except stratified by antibody set and polyQ length, for striatum (left), cortex (middle), and cerebellum (right).

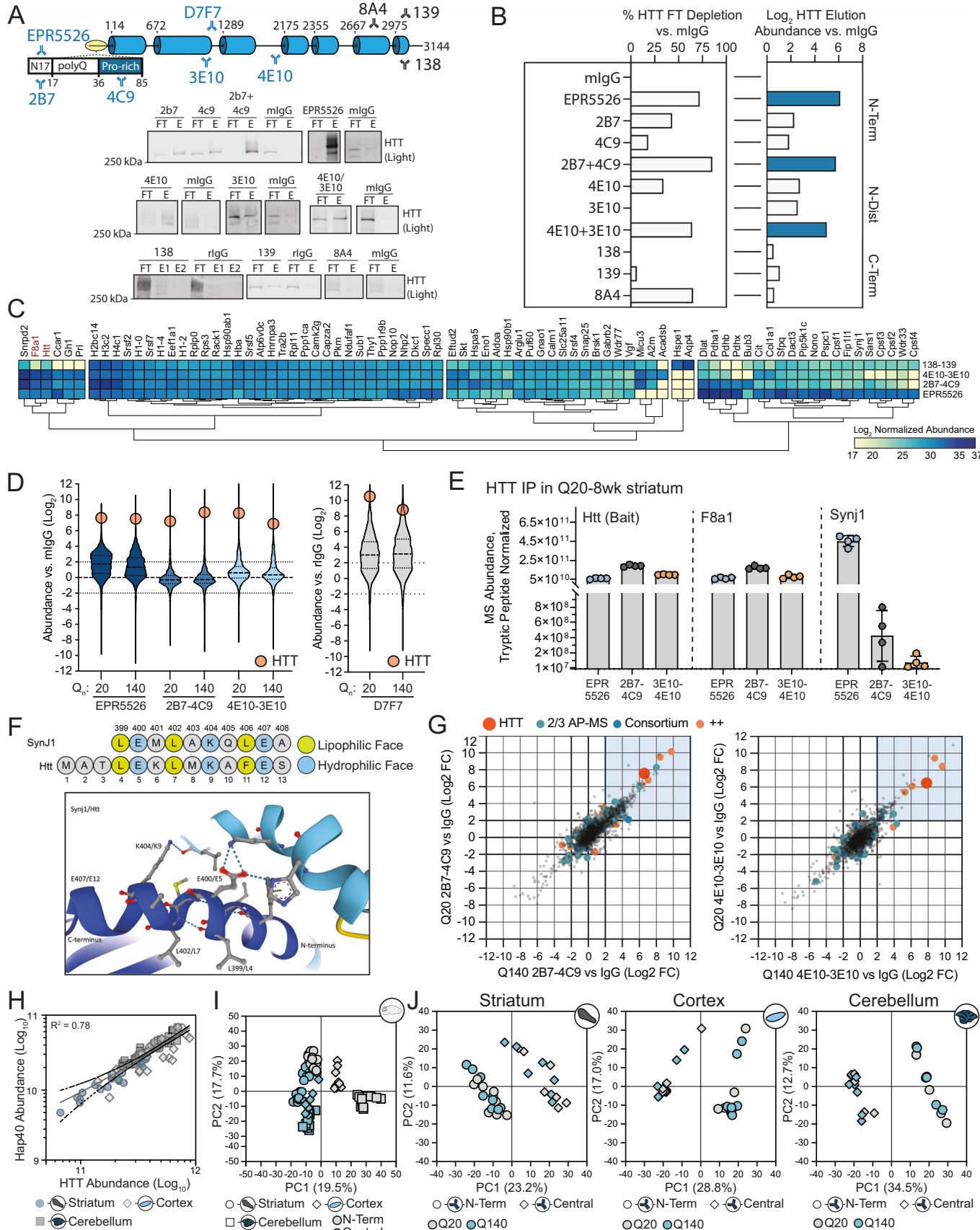

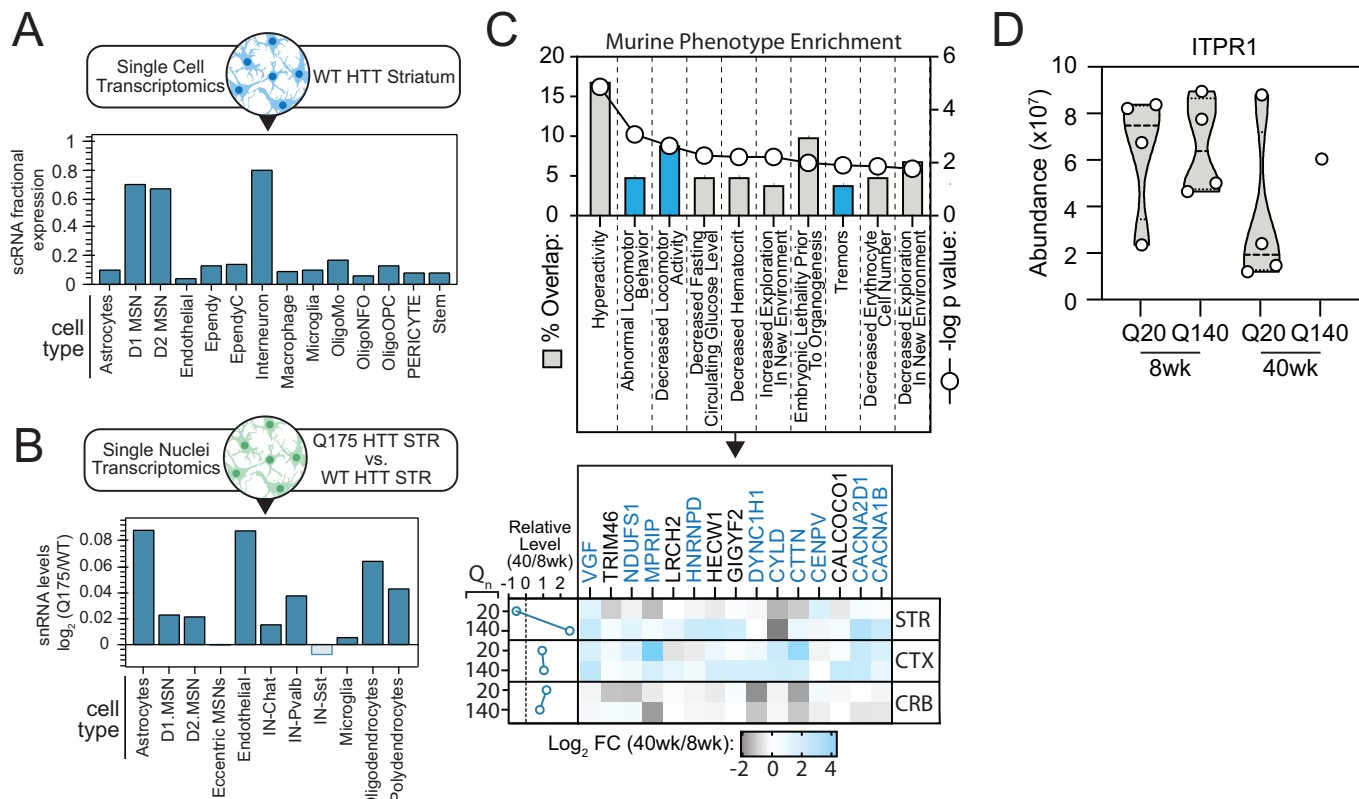

**Figure EV2. Multiomics and functional ontology analysis of striatum-enriched HIP genetic modifiers.**

(A, B) Preferential single cell type RNA fractional expression (A) and mHTT-induced single nuclei RNA levels (B) of proteins from STRING network in Fig. 7D. (C) Functional overrepresentation of murine phenotypes from proteins in Fig. 7D (top) and the corresponding annotated proteins with mHTT-dependent changes in interaction levels in the striatum, cortex, and cerebellum. Over-represented pathways were determined by EnrichR (https://maayanlab.cloud/Enrichr/) using a Fisher's Exact test and corrected for multiple hypothesis testing using the Benjamini-Hochberg method. (D) Interaction levels of ITRP1 with HTT determined by quantitative IP-MS. Individual replicates ($n = 4$) and median protein abundance values are shown.

