## [Peer Review File · Molecular Systems Biology]

Multi-epitope immunocapture of huntingtin reveals striatum-selective molecular signatures

Joshua Justice, Todd Greco, Josiah Hutton, Tavis Reed, Megan Mair, Juan Botas, and Ileana Cristea

Corresponding author(s): Ileana Cristea (icristea@princeton.edu)

Review Timeline:

Submission Date:	7th Sep 24
Editorial Decision:	28th Oct 24
Revision Received:	22nd Dec 24
Editorial Decision:	14th Feb 25
Revision Received:	3rd Mar 25
Accepted:	13th Mar 25

Editor: Poonam Bheda

Transaction Report:

28th Oct 2024

Manuscript Number: MSB-2024-12595

Title: Multi-epitope immunocapture of huntingtin reveals striatum-selective molecular signatures

Dear Prof Cristea,

Thank you for the submission of your manuscript to Molecular Systems Biology. We have now received feedback from the three reviewers who agreed to evaluate your manuscript. As you will see from the reports below, the referees acknowledge the interest of the study and are overall supporting publication of your work pending appropriate revisions.

I think that the recommendations of the reviewers are rather clear and I therefore do not see the need to repeat the comments listed below. All issues raised would need to be satisfactorily addressed. Please let me know in case you would like to discuss in further detail any of the issues raised, I would be happy to schedule a call.

We require:

- 1) A .docx formatted version of the manuscript text (including legends for main figures, EV figures and tables). Please make sure that the changes are highlighted to be clearly visible. Alternatively you may choose to submit your manuscript as a LaTeX file.
- 2) Individual production quality figure files as .eps, .tif, .jpg (one file per figure). For guidance, download the 'Figure Guide PDF' (<https://www.embopress.org/page/journal/17574684/authorguide#figureformat>).
- 3) At EMBO Press we ask authors to provide source data for the main figures. Our source data coordinator will contact you to discuss which figure panels we would need source data for and will also provide you with helpful tips on how to upload and organize the files.
- 4) A .docx formatted letter INCLUDING the reviewers' reports and your detailed point-by-point responses to their comments. As part of the EMBO Press transparent editorial process, the point-by-point response is part of the Peer Review File (PRF), which will be published alongside your paper.
- 5) A complete author checklist, which you can download from our author guidelines (<https://www.embopress.org/page/journal/17574684/authorguide#submissionofrevisions>). Please insert information in the checklist that is also reflected in the manuscript. The completed author checklist will also be part of the PRF.
- 6) Please note that all corresponding authors are required to supply an ORCID ID for their name upon submission of a revised manuscript.
- 7) It is mandatory to include a 'Data Availability' section after the Materials and Methods. Before submitting your revision, primary datasets produced in this study need to be deposited in an appropriate public database, and the accession numbers and database listed under 'Data Availability'. Please remember to provide a reviewer password if the datasets are not yet public (see <https://www.embopress.org/page/journal/17574684/authorguide#dataavailability>).

In case you have no data that requires deposition in a public database, please state so in this section. Note that the Data Availability Section is restricted to new primary data that are part of this study. This study includes no data deposited in external repositories.

- 8) All Materials and Methods need to be described in the main text using our 'Structured Methods' format, which is required for all research articles. According to this format, the Methods section includes a Reagents and Tools Table (listing key reagents, experimental models, software and relevant equipment and including their sources and relevant identifiers) followed by a Methods and Protocols section describing the methods using a step-by-step protocol format. The aim is to facilitate adoption of the methodologies across labs. Please upload the Reagents and Tools table as a separate document when submitting your revised manuscript. More information on how to adhere to this format as well as a downloadable template (.docx) for the Reagents and Tools Table can be found in our author guidelines: <https://www.embopress.org/page/journal/17444292/authorguide#structuredmethods>

An example of a Method paper with Structured Methods can be found here: <https://www.embopress.org/doi/10.15252/msb.20178071>.

9) For data quantification: please specify the name of the statistical test used to generate error bars and P values, the number (n) of independent experiments (specify technical or biological replicates) underlying each data point and the test used to calculate p-values in each figure legend. The figure legends should contain a basic description of n, P and the test applied. Graphs must include a description of the bars and the error bars (s.d., s.e.m.). Please provide exact p values.

10) Our journal encourages inclusion of *data citations in the reference list* to directly cite datasets that were re-used and obtained from public databases. Data citations in the article text are distinct from normal bibliographical citations and should directly link to the database records from which the data can be accessed. In the main text, data citations are formatted as follows: "Data ref: Smith et al, 2001" or "Data ref: NCBI Sequence Read Archive PRJNA342805, 2017". In the Reference list, data citations must be labeled with "[DATASET]". A data reference must provide the database name, accession number/identifiers and a resolvable link to the landing page from which the data can be accessed at the end of the reference. Further instructions are available at .

11) We replaced Supplementary Information with Expanded View (EV) Figures and Tables that are collapsible/expandable online. A maximum of 5 EV Figures can be typeset. EV Figures should be cited as 'Figure EV1, Figure EV2' etc... in the text and their respective legends should be included in the main text after the legends of regular figures.

<https://www.embopress.org/page/journal/17574684/authorguide#expandedview>

13) Author contributions: CRediT has replaced the traditional author contributions section because it offers a systematic machine readable author contributions format that allows for more effective research assessment. Please remove the Authors Contributions from the manuscript and use the free text boxes beneath each contributing author's name in our system to add specific details on the author's contribution. More information is available in our guide to authors.

Please also suggest a striking image or visual abstract to illustrate your article as a PNG file 550 px wide x 300-600 px high. Share synopsis text and image, as well as eTOC:

Please note that these would be the final versions and changes during proofing are usually not allowed

16) As part of the EMBO Publications transparent editorial process initiative (see our policy here:

https://www.embopress.org/transparent-process#Review_Process), Molecular Systems Biology will publish online a Peer Review File (PRF) to accompany accepted manuscripts.

In the event of acceptance, this file will be published in conjunction with your paper and will include the anonymous referee reports, your point-by-point response and all pertinent correspondence relating to the manuscript. Let us know whether you agree with the publication of the PRF and as here, if you want to remove or not any figures from it prior to publication.

Please note that the Authors checklist will be published at the end of the PRF.

Molecular Systems Biology has a "scooping protection" policy, whereby similar findings that are published by others during review or revision are not a criterion for rejection. Should you decide to submit a revised version, I do ask that you get in touch after three months if you have not completed it, to update us on the status.

I look forward to receiving your revised manuscript.

Yours sincerely,

Poonam Bheda, PhD
Scientific Editor
Molecular Systems Biology

Reviewer #1:

Understanding how huntingtin interacting proteins (HIPs) may drive HD pathogenesis in specific brain areas or cell types and how this mechanism may be relevant to the modification of disease progression in models and human patients is very important as this may expand the target space for HD. This question is also of major interest for the study of other diseases. In this study, the authors directly and thoroughly address this question using a wide range of approaches and HD models. They identified new HIPs or further characterize existing HIPs that, in the striatum, may be associated to HD pathogenesis.

This study is highly comprehensive and convincing, providing new insights into neuronal vulnerability to HD, also providing new resources (striatal HIP catalogs). The manuscript is easy to read and the data are well presented and well discussed. Overall, the manuscript contains a lot of new information thanks to the use of several model systems to characterize new HIPs in addition to extensively screening for them.

It could be interesting to discuss the findings on the mediator complex in the context of preprints on genetic modifiers of HD.

Reviewer #2:

In this study, Justice and colleagues aimed to dissect the protein interaction landscapes of wild-type (wtHTT) and mutant HTT (mHTT) to better understand the differences in their interaction profiles. They also explored how these differences manifest in distinct regions of the brains of HD mouse models, namely the striatum, cortex, and cerebellum, at two critical time points: 8 and 40 weeks of age. These time points were chosen to reflect the progressive nature of HD, which typically manifests with adult-onset in patients. A key innovation of this study is the use of a multi-epitope IP-MS approach, employing antibodies targeting different regions of HTT (N-terminus and central), along with the examination of the endogenous, full-length HTT proteoform. Together, this approach allows for a more comprehensive analysis.

The main findings of the manuscript reveal that both wtHTT and mHTT form tissue-specific protein interactions, which could provide an explanation for the region-specific dysfunction observed in HD. Additionally, they found that targeting different epitopes of HTT revealed distinct interaction profiles, underscoring the complexity of HTT's expansive interaction network. Notably, the authors identified 399 novel HTT-interacting proteins and highlighted an interaction between HTT and the Mediator complex, a key player in transcriptional regulation, and a recently identified HD modifier in a GWAS recently shared with the community as a preprint. The authors' findings offer valuable insights into the differential protein interactions across various brain regions, as well as the altered localization and interaction patterns of wt HTT and mHTT, providing a deeper understanding of the molecular underpinnings of HD.

Major points

- The antibody validation in the current manuscript draft is weak (Figure S1). Whilst the authors do show target engagement of their antibodies and evaluate the efficacy of each antibody in an IP, there is no effort made to demonstrate that these antibodies are selective for HTT, a western blot only shows you what you probe for. Given the wealth of MS data collected in this study, the authors could mine this data to demonstrate the selective binding of their chosen antibody set and should employ the gold standard for antibody validation by testing these antibodies in a HTT knock out/knock down control system. Given the availability of HTT lowering agents and KO lines, this is a tractable experiment that would further convince the reader these findings are real.
- In the PCA presented in Figure 1E, there is a clear separation of the data as per different HTT epitopes. However, the authors also suggest a polyQ-dependent effect (grey vs. blue), though this separation is not as clear. In order to make this statement, the authors should clearly detail the analysis by which they derived the polyQ-dependent effect. Further, as the authors only look at 2 polyQ lengths, it is hard to make such a claim without intermediate allele lengths, rather than the WT vs HD mouse models differ, which could be due to a host of other factors caused downstream of the polyQ expansion mutation itself.
- The authors performed a fractionation experiment to investigate how HTT interactions differ between nuclear and cytoplasmic HTT. However, this experiment is unconvincing to the reader as no markers were provided to confirm the successful separation of the nucleus and cytoplasm (e.g. histone vs gapdh). It would be strongly recommended that they include this data to ensure that complete fractionation was achieved.

Minor points

- One of the manuscript's key findings is that the antibody epitope plays a crucial role in distinguishing different protein interactions. However, none of the C-terminal antibodies showed significant efficacy in immunoprecipitating HTT from the tissues, and this issue was not addressed in the paper. Could there be any explanation for the failure of those antibodies in immunoprecipitating HTT? Perhaps the authors could further comment on this in their discussion.
- In the second-to-last paragraph of the discussion, the sentence that begins with 'Additionally, given the...' is incomplete and lacks an ending.

Other comments:

The authors did an excellent job with the presentation and style. Specifically, the use of color-coded figures and schematics enhances the paper's clarity and makes the complex information easier to follow.

Reviewer #3:

The manuscript "Multi-epitope immunocapture of huntingtin reveals striatum-selective molecular signatures" is a scholarly and interesting study that delves deeply into the HTT protein interactions. The studies have broad implications for neurological diseases because regional vulnerability for genetic diseases are likely related to the disease protein interactome in the region/cell types driving the disease. The study is rigorous and informative. The authors use multi-epitope protein interaction studies, subcellular fractionation, thermal proteome profiling, and genetic modifier assays to characterize tissue-specific and epitope-enriched protein interactions of endogenous wild-type and mutant HTT. They focus on the mediator complex (20/46) as an important HTT interactor. The potential nomination of HTT interactors to test as modifiers may be significant.

Point 1. Given the 30 years of molecular investigation into HD by the field, I think this statement should be revised because it is untrue. "Although discovered as the etiological agent of Huntington's disease (HD) in the early 1990's (MacDonald et al, 1993), the molecular mechanisms that link polyglutamine (polyQ) expansion in the N-terminal region of HTT to HD remain unknown". Even for the tissue-specific mechanisms there is a body of work on this.

Point 2. "One hypothesis is that high value disease-relevant candidates are proteins that interact with HTT in the striatum and are linked to HD pathogenesis and/or disease progression." For this statement please put the references that have proposed and validated this. There are many. If this is the case, I would note which human genetic modifiers of HD are HTT interactors. One relevant reference is PMID 17500595.

Point 3. The introduction is very scholarly and lays out how the HTT interactome is quite complex. Please note that the specificity of antibodies to HTT and the defining the HTT interactome in mouse models may contribute to issues with disease relevance and identification of important interactions. I would suggest a brief statement in discussion of the limitations of the studies with HD mouse models to define the HTT interactome.

Point 4. Immunoprecipitation conditions for the experiments contain CaCl₂ and 1% Triton. Perhaps it is worth discussing why this condition was selected in the methods compared to other studies...

Point 5. Briefly define the SAINT specificity filtering and why it is useful.

Point 6. Figure 1. The authors note HTT abundance using the different HTT antibodies could be due to the affinity to wild-type vs. mutant HTT and carefully employ corrections. Maybe also state the tagging HTT can change its localization/interaction if this is the case. This might be apparent by analyzing and comparing to the proteome/HTT interactome defined in previous publications. Did they compare to these data sets?

Point 7. The supplemental figure showing the use of different antibodies to IP HTT is very nice. Are the results in panel B quantification from the western blot or mass spectrometry? Did they compare?

Point 8. Figure 2I. Please define what the colors in the grey rectangle are in the figure legend more clearly. I see it is the level of expression in each genotype/region/age. F8A1 (HAP1) is a highly validated interactor but the expression per tissue/genotype in main figure. Perhaps move from supplemental to main figure.

Point 9. "This is supported by recent genetic findings that transcriptional regulation is differentially impacted in the striatum in HD (Gu et al, 2022; Kumar et al, 2014)." Remove word recently.

Point 10. The interaction of Mediator and HTT should be further validated with a reverse IP with mediator antibodies to the subunits. Also colocalization via immunofluorescence would strengthen the findings.

Point 11. "TCERG1 has been reported as an HTT interaction in yeast two-hybrid studies (Holbert et al, 2001) but had not been

previously confirmed as interacting in mammalian models. It is one of the only genetic loci to be identified as an HD age of onset modifier in multiple human studies (Lee et al, 2019; Lobanov et al, 2022; Andresen et al, 2007)." This sentence may need clarification because they are a number of HD age of onset modifiers validated across multiple human studies.

Point 12. I am surprised most of the HTT interacting proteins do not change in expression for HD and aging.

Point 13. I would put the link for HTT-OMNI website in reference.

Point 14. I strongly urge the authors to edit the manuscript focusing on key points. In the results section there is extensive discussion and perhaps considering removing any redundant points with the actual discussion.

Response to Reviewers

We sincerely thank the editor and the reviewers for the careful assessment of our manuscript. All the comments were constructive and helpful for further enhancing the impact of our manuscript. **We addressed all the reviewers' comments by performing the suggested additional experiments and analyses, as well as all the suggested revisions to the text and figures.** Importantly, these new experiments fully support our initial interpretations and conclusions, while also extending the relevance of our study. Our new analyses resulted in the addition of three new panels in Figure S1 to show antibody specificity assessments. We have also provided these new data in a Supplemental Table and Data. We have revised the text to add key details to the results, and streamlined the results and discussion to focus on the manuscript's main findings and improve readability. Overall, we feel these changes have resulted in a stronger manuscript that demonstrates the rigor and disease relevance of our findings. Below we include our point-by-point responses to each of the specific comments. Our responses are marked with ">>Response:". For convenience, the revised portions of the manuscript text have been included in the point-by-point responses and marked-up in the manuscript file to highlight the changes.

Specific Reviewer Comments

Reviewer #1:

Understanding how huntingtin interacting proteins (HIPs) may drive HD pathogenesis in specific brain areas or cell types and how this mechanism may be relevant to the modification of disease progression in models and human patients is very important as this may expand the target space for HD. This question is also of major interest for the study of other diseases. In this study, the authors directly and thoroughly address this question using a wide range of approaches and HD models. They identified new HIPs or further characterize existing HIPs that, in the striatum, may be associated to HD pathogenesis.

This study is highly comprehensive and convincing, providing new insights into neuronal vulnerability to HD, also providing new resources (striatal HIP catalogs). The manuscript is easy to read and the data are well presented and well discussed. Overall, the manuscript contains a lot of new information thanks to the use of several model systems to characterize new HIPs in addition to extensively screening for them.

It could be interesting to discuss the findings on the mediator complex in the context of preprints on genetic modifiers of HD.

>>Response: We appreciate the Reviewer's positive comments on our study's findings of tissue-specific disease relevant HTT interacting proteins. As suggested by the reviewer, we have expanded our discussion of the disease relevance of mediator interactions given the pre-print report documenting a hastening allele within the MED15 gene that reached genome wide significance in HD individuals. We have put our findings in mouse HD models in the context of the human GWAS (pg 16).

Revised text: “Our data pointing to Med15 as a mHTT effector in mice may reflect a conserved pathological mechanism in humans. The disease hastening MED15 allele in HD individuals, rs177425, was shown to be in linkage disequilibrium with the longer, 13 CAG MED15 allele, suggesting involvement of its polyQ repeat in the modifier effect (Consortium *et al*, 2024), which may facilitate a gain of toxic interaction with mutant HTT’s expanded polyQ.”

Reviewer #2:

In this study, Justice and colleagues aimed to dissect the protein interaction landscapes of wild-type (wtHTT) and mutant HTT (mHTT) to better understand the differences in their interaction profiles. They also explored how these differences manifest in distinct regions of the brains of HD mouse models, namely the striatum, cortex, and cerebellum, at two critical time points: 8 and 40 weeks of age. These time points were chosen to reflect the progressive nature of HD, which typically manifests with adult-onset in patients. A key innovation of this study is the use of a multi-epitope IP-MS approach, employing antibodies targeting different regions of HTT (N-terminus and central), along with the examination of the endogenous, full-length HTT proteoform. Together, this approach allows for a more comprehensive analysis.

The main findings of the manuscript reveal that both wtHTT and mHTT form tissue-specific protein interactions, which could provide an explanation for the region-specific dysfunction observed in HD. Additionally, they found that targeting different epitopes of HTT revealed distinct interaction profiles, underscoring the complexity of HTT's expansive interaction network. Notably, the authors identified 399 novel HTT-interacting proteins and highlighted an interaction between HTT and the Mediator complex, a key player in transcriptional regulation, and a recently identified HD modifier in a GWAS recently shared with the community as a preprint. The authors findings offer valuable insights into the differential protein interactions across various brain regions, as well as the altered localization and interaction patterns of wt HTT and mHTT, providing a deeper understanding of the molecular underpinnings of HD.

>>Response: We appreciate the Reviewer’s encouraging comments on the value of our study’s discoveries for understanding the molecular underpinnings of HD.

Major points

- The antibody validation in the current manuscript draft is weak (Figure S1). Whilst the authors do show target engagement of their antibodies and evaluate the efficacy of each antibody in an IP, there is no effort made to demonstrate that these antibodies are selective for HTT, a western blot only shows you what you probe for. Given the wealth of MS data collected in this study, the authors could mine this data to demonstrate the selective binding of their chosen antibody set and should employ the gold standard for antibody validation by testing these antibodies in a HTT knock out/knock down control system. Given the availability of HTT lowering agents and KO lines, this is a tractable experiment that would further convince the reader these findings are real.

>>Response: We appreciate the reviewer’s comment. Actually, the assessment of antibody specificity was something that we carefully considered at the early stages of our study. We took advantage of our use of multiple HTT antibodies that covered epitopes spanning the N- to C-terminus to assess specificity. These assessments, which we did not include initially in our manuscript in order to streamline the already complex manuscript, allowed us to select the

antibodies used for our analyses. Specifically, in addition to the ability of the antibodies to enrich for HTT, we also evaluated off-target capture by screening for proteins that were (1) co-isolated by only one antibody and (2) present at an abundance similar to or greater than HTT. We have now revised Figure S1 to include a systematic analysis of antibody specificity. In summary, we found that (1) the C-terminal antibodies, which had poor HTT enrichment, likely had non-specific capture of either aquaporin-4 or Hsp10 and (2) while the remaining antibodies show good selectivity for HTT, the EPR5526 antibody exhibited off-target capture, with the leading candidate being synaptojanin-1 (Synj1). Our further analysis supported this observation by identifying a predicted conformational epitope of Synj1 that is conserved within the EPR5526 epitope region of HTT (N17). Hence, these experiments allowed us to exclude the C-terminal antibodies and EPR5526 antibody for subsequent IP-MS analyses and focus on the antibodies with good specificity. We have constructed new figure panels in Fig S1 to show these analyses and revised the first paragraph of the results section (see text below).

Revised text: Next, we evaluated antibody specificity using immunoaffinity purification paired with tandem mass spectrometry (IP-MS). Consistent with the western blot analysis, the C-terminal antibodies 138-139 showed ~50-100x lower capture of HTT. The highest abundance captured proteins were aquaporin-4 (Aqp4) and Hsp10 (Hspe1), which were uniquely identified in the 138-139 IPs (Fig S1C), suggesting potential off-target capture. For the remaining isolations with EPR5526, 2B7-4C9, 4E10-3E10, the intended bait, HTT, and its known interacting partner, HAP40 (F8a1) were consistently isolated and among the most abundant captured proteins (Fig S1C). However, we observed that EPR5526 captured a unique population of interacting proteins compared to 2B7-4C9 and 4E10-3E10 (Fig S1C & 1D). We investigated this further in the striatum and found that, despite the efficient capture of HTT/HAP40 in an ~1:1 ratio, EPR5526 also captured synaptojanin-1 (Synj1) with ~10x higher abundance than HTT/HAP40 (Fig S1E). Interestingly, the capture may be based on a shared conformational epitope. The primary sequence alignment of HTT-Synj1 and the Synj1 crystal structure (AlphaFold prediction) supported an alpha-helix conformation (Fig S1F), but we did not observe an anti-EPR5526 immunoreactive band in the expected region (~173 kDa) by western blotting. Overall, based on our IP efficiency and specificity analyses, we proceeded with 2B7-4C9 and 4E10-3E10 antibodies for the complete workflow (Fig 1A). Both of these antibody combinations showed effective capture of HTT relative to the isotype matched controls for Q20 and Q140 genotypes from whole brain (Fig S1C) and striatum (Fig S1D & 1G), as well as the enrichment for validated HTT interacting proteins (Fig S1G), including HAP40.

- In the PCA presented in Figure 1E, there is a clear separation of the data as per different HTT epitopes. However, the authors also suggest a polyQ-dependent effect (grey vs. blue), though this separation is not as clear. In order to make this statement, the authors should clearly detail the analysis by which they derived the polyQ-dependent effect. Further, as the authors only look at 2 polyQ lengths, it is hard to make such a claim without intermediate allele lengths, rather that the WT vs HD mouse models differ, which could be due to a host of other factors caused downstream of the polyQ expansion mutation itself.

>>**Response:** We agree with the reviewer that strong claims of polyQ-dependence should be avoided based on PCA analysis. We have modified the sentence that described Fig 1E to include additional details that more accurately describe the data shown.

Revised text: “Moreover, when considering the abundance profiles of SAINT-filtered HTT interacting partners isolated from the striatum and cortex, the Q140 40-week-old mice were more distinct when using HTT antibodies targeting the N-terminal versus the Central epitope, as shown by this sample group’s variance in PC1 (Figure 1E, blue vs. grey). In contrast, IP samples in the cerebellum showed the most separation by the antibody variable, but less so by the mHTT or age variables (Figure 1E, diamonds vs circles). Overall, these results suggest that different subsets of proteins interacting with wild-type or mutant HTT may cause preferential immunocapture depending on the targeted epitopes.”

-The authors performed a fractionation experiment to investigate how HTT interactions differ between nuclear and cytoplasmic HTT. However, this experiment is unconvincing to the reader as no markers were provided to confirm the successful separation of the nucleus and cytoplasm (e.g. histone vs gapdh). It would be strongly recommended that they include this data to ensure that complete fractionation was achieved.

>>**Response:** We agree that assessing the fractionation performance is important. Indeed, we have monitored such markers and presented a panel of subcellular localization markers in Figure 4F, including histones, lamins, HNRNPs, ribosomes (among others). In our effort to be concise, we regret that our original text did not clearly explain that the Nuclear/Cytoplasm ratios of these protein markers were measured. In summary, this figure shows that nuclear markers (histones and lamins) are on average ≥ 4 -fold enriched in the nuclear fraction, while ribosomal and STAT proteins, typical in the cytoplasm, were ≥ 4 -fold enriched in the cytoplasmic fraction. We have modified the text accordingly.

Revised text: “As expected, HTT was found predominantly in the cytosolic fraction, which was enriched in known cytoplasmic markers (Fig 4F). HTT was also detectable at low levels in the nucleus enriched fraction, which showed ≥ 4 -fold enrichment of histone and other nuclear markers (Fig 4F)”

Minor points

-One of the manuscript's key findings is that the antibody epitope plays a crucial role in distinguishing different protein interactions. However, none of the C-terminal antibodies showed significant efficacy in immunoprecipitating HTT from the tissues, and this issue was not addressed in the paper. Could there be any explanation for the failure of those antibodies in immunoprecipitating HTT? Perhaps the authors could further comment on this in their discussion.

>>**Response:** While we do not have a clear explanation for the poor performance, we now describe the C-terminal antibody performance in the first paragraph of the Results, and provide a comment on the possible underlying cause in the Discussion. Additionally, we have performed additional analyses comparing antibody capture, which include the above-mentioned C-terminal

epitope HTT antibodies, 138 and 139. As shown in new panels of Figure EV1, our assessment of the C-terminal antibodies (138 & 139) showed that these antibodies captured ~50 - 100x less HTT than the other HTT antibodies, and other known interactions were reduced. In addition, we found that they likely have off-target capture of aquaporin-4. Hence, these antibodies were not selected for our next analyses. Anecdotally, attempts by HD researchers to generate functional affinity-tagged HTT usually encounter greater challenges at the C-terminus versus the N-terminus, suggesting the C-terminus is less structurally accessible/accommodating; perhaps this also impacts antibody accessibility.

Revised text (Results): “Next, we evaluated antibody specificity using immunoaffinity purification paired with tandem mass spectrometry (IP-MS) (Table EV1). Consistent with the western blot analysis, C-terminal antibodies 138-139 showed ~50-100x lower capture of HTT. The highest abundance captured proteins were aquaporin-4 (Aqp4) and Hsp10 (Hspe1), which were uniquely identified in the 138-139 IPs (Fig EV1C and Table EV1), suggesting potential off-target capture. For the remaining isolations with EPR5526, 2B7-4C9, 4E10-3E10, the intended bait, HTT, and its known interacting partner, HAP40 (F8a1) were consistently isolated and among the most abundant captured proteins (Fig EV1C). However, we observed that EPR5526 captured a unique population of interacting proteins compared to 2B7-4C9 and 4E10-3E10 (Fig EV1C & 1D). We investigated this further in the striatum and found that, despite the efficient capture of HTT/HAP40 in an ~1:1 ratio, EPR5526 also captured synaptojanin-1 (Synj1) with ~10x higher abundance than HTT/HAP40 (Fig EV1E). Interestingly, the capture may be based on a shared conformational epitope. The primary sequence alignment of HTT-Synj1 and the Synj1 crystal structure (AlphaFold prediction) supported an alpha-helix conformation (Fig EV1F), but we did not observe an anti-EPR5526 immunoreactive band in the expected region (~173 kDa) by western blotting. Overall, based on our IP efficiency and specificity analyses, we proceeded with 2B7-4C9 and 4E10-3E10 antibodies for the complete workflow (Fig 1A). Both of these antibody combinations showed effective capture of HTT relative to the isotype matched controls for Q20 and Q140 genotypes from whole brain (Fig EV1C) and striatum (Fig EV1D & 1G), as well as the enrichment for validated HTT interacting proteins (Fig EV1G), including HAP40”

Revised text (Discussion): “With respect to the C-terminus, occurrence of domain-specific interactions or long-distance interactions with the N-terminal domains (Vijayvargia et al, 2016) may occlude antibody epitopes and contribute to the poor performance of C-terminal HTT antibodies.”

- In the second-to-last paragraph of the discussion, the sentence that begins with 'Additionally, given the...' is incomplete and lacks an ending.

>>**Response:** Thank you for finding this typographical error. We deleted these words. Additionally, to address another reviewer’s comment, we revised the conclusion of this paragraph to discuss our findings related to the mediator complex in mouse HD models in the context of the human GWAS.

Revised text: “Our data pointing to Med15 as a mHTT effector in mice may be a conserved pathological mechanism in humans. The disease hastening MED15 allele in HD individuals, rs177425, was shown to be in linkage disequilibrium with the longer, 13 CAG MED15 allele, suggesting involvement of its polyQ repeat in the modifier effect (Consortium *et al*, 2024), which may facilitate a gain of toxic interaction with mutant HTT’s expanded polyQ.”

Other comments:

The authors did an excellent job with the presentation and style. Specifically, the use of color-coded figures and schematics enhances the paper's clarity and makes the complex information easier to follow.

>>**Response:** We appreciate that the reviewer noted our deliberate efforts to construct figures with clear annotations, graphics, and consistent visual styles to improve readability.

Reviewer #3:

The manuscript "Multi-epitope immunocapture of huntingtin reveals striatum-selective molecular signatures" is a scholarly and interesting study that delves deeply into the HTT protein interactions. The studies have broad implications for neurological diseases because regional vulnerability for genetic diseases are likely related to the disease protein interactome in the region/ cell types driving the disease. The study is rigorous and informative. The authors use multi-epitope protein interaction studies, subcellular fractionation, thermal proteome profiling, and genetic modifier assays to characterize tissue-specific and epitope-enriched protein interactions of endogenous wild-type and mutant HTT. They focus on the mediator complex (20/46) as an important HTT interactor. The potential nomination of HTT interactors to test as modifiers may be significant.

Point 1. Given the 30 years of molecular investigation into HD by the field, I think this statement should be revised because it is untrue. "Although discovered as the etiological agent of Huntington's disease (HD) in the early 1990's (MacDonald et al, 1993), the molecular mechanisms that link polyglutamine (polyQ) expansion in the N-terminal region of HTT to HD remain unknown". Even for the tissue-specific mechanisms there is a body of work on this.

>>**Response:** Thank you, we have revised the sentence to avoid implying that there are no mechanistic insights into the how polyQ expansion contributes to HD.

Revised text: "Although discovered as the etiological agent of Huntington's disease (HD) in the early 1990's (MacDonald et al, 1993), the precise mechanisms by which polyglutamine (polyQ) expansion within the N-terminal region of HTT causes toxicity in vulnerable brain cells remains an active area of study."

Point 2. "One hypothesis is that high value disease-relevant candidates are proteins that interact with HTT in the striatum and are linked to HD pathogenesis and/or disease progression." For this statement please put the references that have proposed and validated this. There are many. If this is the case, I would note which human genetic modifiers of HD are HTT interactors. One relevant reference is PMID 17500595.

>>**Response:** Thank you for the suggestion. We have included the relevant references for the statement highlighted. We have also enumerated the handful of human genetic modifiers of HD that are interactions. The revised sentences are as follows:

Revised text: One hypothesis is that high value disease-relevant candidates are proteins that interact with HTT in the striatum and are linked to HD pathogenesis and/or disease progression (Goehler *et al*, 2004; Kaltenbach *et al*, 2007; Neri, 2011; Shirasaki *et al*, 2012; Greco *et al*, 2022). While most HTT protein interactions that are disease modifiers have been defined in HD model organisms, there are several interactions identified as human genetic HD modifiers from genome wide association studies, including TCERG1 (Kaltenbach *et al*, 2007; Andresen *et al*, 2007; Lobanov *et al*, 2022), MSH3 (Shirasaki *et al*, 2012; Flower *et al*, 2019), and more recently MLH1-PMS2 (Lee *et al*, 2017; Sun *et al*, 2024) and POLD1 (Ripaud *et al*, 2014; Consortium *et al*, 2024).

Point 3. The introduction is very scholarly and lays out how the HTT interactome is quite complex. Please note that the specificity of antibodies to HTT and the defining the HTT interactome in mouse models may contribute to issues with disease relevance and identification of important interactions. I would suggest a brief statement in discussion of the limitations of the studies with HD mouse models to define the HTT interactome.

>>**Response:** We have included a few sentences related to limitations of HD mouse models when studying interactions in the first paragraph of Discussion.

Revised text: One consideration is that a little over 1/3 of the HTT interaction knowledgebase is derived from various HD mouse models. While HD mouse models exhibit motor and cognitive impairments that mimic the human disease, there are limitations. Caveats include differences in HTT gene sequences and structure, HD phenotypes, and disease progression of various mouse models (e.g., R6/2, YAC, and BACHD), as well as the inability to recapitulate more complex aspects of the human disease. Therefore, multiple lines of evidence are valuable when interpreting an interaction's potential disease relevance. Towards this goal, we focused on striatum-enriched and polyQ-dependent interaction candidates that produced a reasonable subset of 165 candidates.

Point 4. Immunoprecipitation conditions for the experiments contain CaCl₂ and 1% Triton. Perhaps it is worth discussing why this condition was selected in the methods compared to other studies...

>>**Response:** We have added sentences to the Method section where the lysis buffer was first described, "Magnetic bead conjugation".

Revised Text: "This buffer composition was selected based on its extensive prior assessment and implementation. It was designed to minimize disruption of cellular complexes and provide ion cofactors to maintain various enzymatic activities (Cristea *et al*, 2005; Joshi *et al*, 2013; Mathias *et al*, 2016), and with some modifications, it was previously optimized for the capture of FLAG-HTT complexes (Greco *et al*, 2022). The non-denaturing Triton X-100 detergent provides reasonable protein extraction while maintaining interactions within lipid rafts (Kramer *et al*,

2012; Miteva *et al*, 2013), which is relevant given HTT can be membrane-associated (Kegel *et al*, 2005).” (Cristea *et al*, 2005; Joshi *et al*, 2013)(Kramer *et al*, 2012).”

Point 5. Briefly define the SAINT specificity filtering and why it is useful.

>>**Response:** We have included the following sentence in the third paragraph of the results section.

Revised text: “This computational tool models the MS signal distribution of proteins from the control (IgG) IPs and the experimental (HTT) IPs to define experiment-optimized thresholds for removal of non-specific interactions.”

Point 6. Figure 1. The authors note HTT abundance using the different HTT antibodies could be due to the affinity to wild-type vs. mutant HTT and carefully employ corrections. Maybe also state the tagging HTT can change its localization/interaction if this is the case. This might be apparent by analyzing and comparing to the proteome/HTT interactome defined in previous publications. Did they compare to these data sets?

>>**Response:** This is an interesting point. One of the most common epitope tags for IP studies of HTT is an N-terminal 3xFLAG, which has been used by us and others in knock-in mouse models (PMIDs: 22556411 and 35148841). Although our previous study of FLAG-HTT interactions in HD mice (PMID: 35148841) did not observe obvious differences in the affinity of anti-FLAG to capture wild-type or mutant FLAG-HTT, FLAG-HTT IPs were underrepresented in nuclear annotated proteins, similar to our observations in this study when using HTT antibodies against the N-terminus. While this was an interesting finding, we thought its specific relevance to this study was tangential. However, the broader issue of antibody preference for certain states/localizations of HTT is underappreciated. From that perspective, our study defines these characteristics and leverages the orthogonal results to have a more complete repertoire of HTT interacting partners. We have included a sentence in the discussion summarizing these challenges and providing context to our study.

Revised text: “The broader issue of antibody preference for certain states/localizations of HTT is underappreciated. From this perspective, our study starts to define these epitope-dependent preferences and leverages the orthogonal results to have a more complete repertoire of HTT interacting partners.”

Point 7. The supplemental figure showing the use of different antibodies to IP HTT is very nice. Are the results in panel B quantification from the western blot or mass spectrometry? Did they compare?

>>**Response:** The results in panel B are quantification from western blot, which we clarify in the figure legend. We did not directly compare WB and MS quantification of HTT.

Point 8. Figure 2I. Please define what the colors in the grey rectangle are in the figure legend more clearly. I see it is the level of expression in each genotype/region/age. F8A1 (HAP1) is a highly validated interactor but the expression per tissue/genotype in main figure. Perhaps move

from supplemental to main figure.

>>**Response:** Thank you for the suggestion. We have now improved the description of the node fill and border colors in the figure legend, as detailed below.

Revised text: “Nodes are color-coded to represent the HIP abundance age ratio (40wk/8wk) for each genotype (Q20 and Q140) in the striatum, cortex, and cerebellum to visualize age-dependent interactions. Node border color indicates whether the HIP was captured with the N-term (*grey*) or Central (*blue*) antibodies.

Point 9. "This is supported by recent genetic findings that transcriptional regulation is differentially impacted in the striatum in HD (Gu et al, 2022; Kumar et al, 2014)." Remove word recently.

>>**Response:** The word “recent” has been removed.

Point 10. The interaction of Mediator and HTT should be further validated with a reverse IP with mediator antibodies to the subunits. Also colocalization via immunofluorescence would strengthen the findings.

>>**Response:** We agree that the Mediator interaction is of significant interest. That is why in our manuscript, we also performed intra-complex abundance comparisons of mediator subunits as well as IP-MS validation of the interaction using independent brain samples and individual HTT antibodies, plus a new fifth antibody (Figure 4D). Comparison of subunit relative abundances showed the greatest capture of subunits at the interface of the head and tail domains within the Mediator 3D structure. In the striatum, these “hot-spots” were Med15 and Med27. Moreover, our results from single antibody IP-MS (Fig 4D) shows that mediator subunits, including Med15 and Med27, can be efficiently co-isolated with HTT by different antibodies, either the 4C9 or 3E10 HTT antibodies, and modestly by the 4E10 antibody. Taken together, the structure-informed preferential enrichment of mediator complex subunits by three individual antibodies with non-overlapping epitopes provides solid evidence for HTT’s interaction with Mediator. While reciprocal isolations are indeed valuable, we feel that, in this context, our existing data already validates the association of HTT with Mediator in multiple independent experiments. One additional consideration is that, given our observations, it is likely that the association of HTT with Mediator is not binary and may be facilitated by multiple subunits. We feel that further characterization of the interaction interface and the precise relationship between the subcellular localization of HTT and Med15 and/or Med27 (via microscopy co-localization studies) in relevant cellular HD models are outside the scope of this study and will be valuable for future experiments. We have revised the Discussion to highlight the likelihood of multiple points of interaction between HTT and Mediator subunits within the tail domain, and discuss the functional implications.

Point 11. "TCERG1 has been reported as an HTT interaction in yeast two-hybrid studies (Holbert et al, 2001) but had not been previously confirmed as interacting in mammalian models. It is one of the only genetic loci to be identified as an HD age of onset modifier in multiple human studies (Lee et al, 2019; Lobanov et al, 2022; Andresen et al, 2007)." This sentence may

need clarification because they are a number of HD age of onset modifiers validated across multiple human studies.

>>**Response:** We have removed the sentence: “It is one of the only genetic loci to be identified as an HD age of onset modifier in multiple human studies (Lee et al, 2019; Lobanov et al, 2022; Andresen et al, 2007)”, to be consistent with the additional description of genetic modifiers x PPIs that we added as part of our reply to point 2 above.

Point 12. I am surprised most of the HTT interacting proteins do not change in expression for HD and aging.

>>**Response:** We agree, this is an intriguing result (RE: Figure 6E). Interestingly, we first reported a similar observation (lack of overlap) between differential HTT interactions vs. the corresponding proteome abundance in FLAG-tagged Q20 and Q140 knock-in mice (PMID: 35148841). In the current study (Figure 6E), the degree of proteome dysregulation in Q140 mice we observe is consistent with proteomics analyses performed in an allelic polyQ series of HD mice, which showed ~500 total differential proteins (PMID: 26900923, Figure 7b). We have added a sentence in the Results noting our prior observation of this relationship between the HD proteome and HTT interactome in a similar mouse HD model.

Point 13. I would put the link for HTT-OMNI website in reference.

>>**Response:** Reference edited.

Point 14. I strongly urge the authors to edit the manuscript focusing on key points. In the results section there is extensive discussion and perhaps considering removing any redundant points with the actual discussion.

>>**Response:** We thank the reviewer for the constructive suggestion; we have checked the manuscript to avoid redundancy.

14th Feb 2025

Manuscript Number: MSB-2024-12595R

Title: Multi-epitope immunocapture of huntingtin reveals striatum-selective molecular signatures

Dear Prof Cristea,

Thank you for the submission of your revised manuscript to Molecular Systems Biology. We have now received the enclosed reports from the referees that were asked to re-assess it. As you will see the reviewers are now globally supportive and I am pleased to inform you that we will be able to accept your manuscript pending the following final amendments:

1) Please format the Data availability section according to the example below (including links to the direct dataset and not just the repository) and ensure that all datasets are now publicly released:

"The datasets and computer code produced in this study are available in the following databases:

- Chip-Seq data: Gene Expression Omnibus GSE46748 (<https://www.ncbi.nlm.nih.gov/geo/query/acc.cgi?acc=GSE46748>)
- Modeling computer scripts: GitHub (<https://github.com/SysBioChalmers/GECKO/releases/tag/v1.0>)
- [data type]: [full name of the resource] [accession number/identifier] ([doi or URL or identifiers.org/DATABASE:ACCESSION])"

2) Our journal encourages inclusion of *data citations in the reference list* to directly cite datasets that were re-used and obtained from public databases. Data citations in the article text are distinct from normal bibliographical citations and should directly link to the database records from which the data can be accessed. In the main text, data citations are formatted as follows: "Data ref: Smith et al, 2001" or "Data ref: NCBI Sequence Read Archive PRJNA342805, 2017". In the Reference list, data citations must be labeled with "[DATASET]". A data reference must provide the database name, accession number/identifiers and a resolvable link to the landing page from which the data can be accessed at the end of the reference. Further instructions are available at .

3) Data not shown: We do not allow statements/conclusions with "data not shown" as per our guidelines on "Unpublished Data". All data referred to in the paper should be displayed in the main or Expanded View figures. Please remove from page 9.

4) In the Methods, please take care of the following:

- Animals: Please ensure that an ethics statement and the approval committee for research on animals is included in the section where animal experiments are described in the Methods. Please also ensure that housing conditions as well as gender, age, and origin of the animals involved in experiments is reported.
- Cell lines: Although you have indicated in the Author Checklist that information on authentication and testing for mycoplasma contamination of the cell lines has been included, this information appears to be missing in the Methods in the main manuscript file.

5) All Materials and Methods need to be described in the main text using our 'Structured Methods' format. According to this format, the Methods section includes a Reagents and Tools Table (listing key reagents, experimental models, software and relevant equipment and including their sources and relevant identifiers) followed by a Methods and Protocols section describing the methods, ideally using a step-by-step protocol format. The aim is to facilitate adoption of the methodologies across labs. Please download and fill our Reagents and Tools Table template (.docx), which you can find in our author guidelines: <https://www.embopress.org/page/journal/14693178/authorguide#structuredmethods>.

<https://www.embopress.org/doi/10.15252/msb.20178071>. "

6) Please place individual sections of the manuscript in the following order: Title page - Abstract & Keywords - Introduction - Results - Discussion - Methods - Data Availability - Acknowledgements - Disclosure and Competing Interests Statement - References - Figure Legends - Expanded View Figure Legends.

7) For the figures and figure legends, please take care of the following:

- There are callouts for Fig. EV5 and EV6 in Expanded View Table Legends - please check as these appear to be typos since there are no Fig. EV5 and EV6 in your submission
- For the files uploaded as Dataset EV1-6 - Dataset EV3 should not be a dataset, but rather a table, please rename this as Table EV1 and upload it as Expanded View content with its legend included above the table in the Excel file. For the remaining Dataset files, please rename them to Dataset EV1-5, and ensure that the file names are correct, as well as their callouts throughout the manuscript. The Expanded View Table legends should be removed from the main manuscript and assuming these are the legends for the Dataset EV files, please include them within each Dataset file as a separate tab/sheet in Excel.
- Please make sure to update the callouts of all figures in the main manuscript text. Currently figure callouts are missing for Fig. 3F-H, 4J, 4N, 7I-J.
- Please note that the exact p values are not provided in the legends of figures 4N, 7I, J.
- Please indicate the statistical test used for data analysis in the legends of figures 2G, H; 5D, 6B, C, E; 7A, B, I, J; EV2 C.

- Please indicate what ** represents; if this represents p value(s), please indicate the statistical test used and where appropriate, specify the exact p value in the legend(s) of figure(s) 4K.
 - Please note that the box plots need to be defined in terms of minima, maxima, centre, bounds of box and whiskers, and percentile in the legends of figures 7I, J.
 - Please note that information related to n is missing in the legends of figures 2A, B, C; 4F, M; 6D, E; 7F, G, H.
 - Please note that the error bars are not defined in the legends of figures 4M, EV1 E.
 - Please note that the scale bar needs to be defined for figure 2K.
- 8) Funding: Please ensure that all funding sources are entered into the manuscript file - currently the following is missing: grant number for CHDI - A-18331; NSF | National Science Foundation Graduate Research Fellowship Program (GRFP) - DGE-2039656
- 9) Synopsis: Please check your synopsis text and image before submission with your revised manuscript. Please be aware that in the proof stage minor corrections only are allowed (e.g., typos).
- 10) Source Data: The Source Data requested in the Source Data checklist was for microscopy image data from Figure 2K. The files that you provided appear to be related to Figure 1, but we are not able to see anything in the tif files - they appear completely black. Please ensure that you upload the appropriate and visible Source Data for Figure 2K organized as a single source data file (zipped) per figure for main figures with the panels clearly visible in the folder structure.
- 11) As part of the EMBO Publications transparent editorial process initiative (see our policy here: https://www.embopress.org/transparent-process#Review_Process), Molecular Systems Biology will publish online a Peer Review File (PRF) to accompany accepted manuscripts. This file will be published in conjunction with your paper and will include the anonymous referee reports, your point-by-point response and all pertinent correspondence relating to the manuscript. Let us know whether you agree with the publication of the PRF and as here, if you want to remove or not any figures from it prior to publication. Please note that the Authors checklist will be published at the end of the PRF.
- 12) After your paper is published, we will promote it on social media. If you have any handles or hashtags for Bluesky or X you would like included, please let us know.
- 13) Please provide a point-by-point letter INCLUDING my comments and your detailed responses (as Word file).

I look forward to reading a new revised version of your manuscript as soon as possible.

Yours sincerely,

Poonam Bheda, PhD
Scientific Editor
Molecular Systems Biology

Reviewer #2:

In this resubmission, the authors have amply and suitably addressed the concerns laid out in the initial review.

Reviewer #3:

The authors have been responsive to my prior review.

1) Please format the Data availability section according to the example below (including links to the direct dataset and not just the repository) and ensure that all datasets are now publicly released:

"The datasets and computer code produced in this study are available in the following databases:

- Chip-Seq data: Gene Expression Omnibus GSE46748
(<https://www.ncbi.nlm.nih.gov/geo/query/acc.cgi?acc=GSE46748>)

- Modeling computer scripts: GitHub
(<https://github.com/SysBioChalmers/GECKO/releases/tag/v1.0>)

- [data type]: [full name of the resource] [accession number/identifier] ([doi or URL or identifiers.org/DATABASE:ACCESSION])"

>>Response: The formatting of the Data Availability section has been updated, and the datasets have been confirmed as public.

2) Our journal encourages inclusion of *data citations in the reference list* to directly cite datasets that were re-used and obtained from public databases. Data citations in the article text are distinct from normal bibliographical citations and should directly link to the database records from which the data can be accessed. In the main text, data citations are formatted as follows: "Data ref: Smith et al, 2001" or "Data ref: NCBI Sequence Read Archive PRJNA342805, 2017". In the Reference list, data citations must be labeled with "[DATASET]". A data reference must provide the database name, accession number/identifiers and a resolvable link to the landing page from which the data can be accessed at the end of the reference. Further instructions are available at <https://www.embopress.org/page/journal/17574684/authorguide#referencesformat>.

>>Response: N/A, as we did not re-use data from public databases.

3) Data not shown: We do not allow statements/conclusions with "data not shown" as per our guidelines on "Unpublished Data". All data referred to in the paper should be displayed in the main or Expanded View figures. Please remove from page 9.

>>Response: We removed the statement from page 9.

4) In the Methods, please take care of the following:

- Animals: Please ensure that an ethics statement and the approval committee for research on animals is included in the section where animal experiments are described in the Methods. Please also ensure that housing conditions as well as gender, age, and origin of the animals involved in experiments is reported.

>> Response: Gender, age and origin of the mice have been stated. The mice were housed and sacrificed by an independent organization, Jackson Labs, so

there is no approval committee; however, I have included a statement that paraphrases their ethics.

- Cell lines: Although you have indicated in the Author Checklist that information on authentication and testing for mycoplasma contamination of the cell lines has been included, this information appears to be missing in the Methods in the main manuscript file.

>>Response: We have added several sentences in the “BE(2)-C cell culture and Crispr Knockout” method section describing our procedure to use mass spectrometry-based proteomics to screen for mycoplasma bacterial contamination.

5) All Materials and Methods need to be described in the main text using our 'Structured Methods' format. According to this format, the Methods section includes a Reagents and Tools Table (listing key reagents, experimental models, software and relevant equipment and including their sources and relevant identifiers) followed by a Methods and Protocols section describing the methods, ideally using a step-by-step protocol format. The aim is to facilitate adoption of the methodologies across labs.

Please download and fill our Reagents and Tools Table template (.docx), which you can find in our author

guidelines: <https://www.embopress.org/page/journal/14693178/authorguide#structuredmethods>.

An example of a Method paper with Structured Methods can be found here: <https://www.embopress.org/doi/10.15252/msb.20178071>. "

>>Response: The Reagents and Tools Table has been prepared according to the template.

6) Please place individual sections of the manuscript in the following order: Title page - Abstract & Keywords - Introduction - Results - Discussion - Methods - Data Availability - Acknowledgements - Disclosure and Competing Interests Statement - References - Figure Legends - Expanded View Figure Legends.

>>Response: The manuscript has been ordered as above.

7) For the figures and figure legends, please take care of the following:

- There are callouts for Fig. EV5 and EV6 in Expanded View Table Legends - please

check as these appear to be typos since there are no Fig. EV5 and EV6 in your submission

- For the files uploaded as Dataset EV1-6 - Dataset EV3 should not be a dataset, but rather a table, please rename this as Table EV1 and upload it as Expanded View content with its legend included above the table in the Excel file. For the remaining Dataset files, please rename them to Dataset EV1-5, and ensure that the file names are correct, as well as their callouts throughout the manuscript. The Expanded View Table legends should be removed from the main manuscript and assuming these are the legends for the Dataset EV files, please include them within each Dataset file as a separate tab/sheet in Excel.
- Please make sure to update the callouts of all figures in the main manuscript text. Currently figure callouts are missing for Fig. 3F-H, 4J, 4N, 7I-J.
- Please note that the exact p values are not provided in the legends of figures 4N, 7I, J.
- Please indicate the statistical test used for data analysis in the legends of figures 2G, H; 5D, 6B, C, E; 7A, B, I, J; EV2 C.
- Please indicate what ** represents; if this represents p value(s), please indicate the statistical test used and where appropriate, specify the exact p value in the legend(s) of figure(s) 4K.
- Please note that the box plots need to be defined in terms of minima, maxima, centre, bounds of box and whiskers, and percentile in the legends of figures 7I, J.
- Please note that information related to n is missing in the legends of figures 2A, B, C; 4F, M; 6D, E; 7F, G, H.
- Please note that the error bars are not defined in the legends of figures 4M, EV1 E.
- Please note that the scale bar needs to be defined for figure 2K.

>>Response: All these subpoints have been addressed in the legends.

8) Funding: Please ensure that all funding sources are entered into the manuscript file - currently the following is missing: grant number for CHDI - A-18331; NSF | National Science Foundation Graduate Research Fellowship Program (GRFP) - DGE-2039656

>>Response: The grant numbers have been placed into the main manuscript file.

9) Synopsis: Please check your synopsis text and image before submission with your revised manuscript. Please be aware that in the proof stage minor corrections only are allowed (e.g., typos).

>>Response: The synopsis image has been checked.

10) Source Data: The Source Data requested in the Source Data checklist was for microscopy image data from Figure 2K. The files that you provided appear to be related to Figure 1, but we are not able to see anything in the tif files - they appear completely black. Please ensure that you upload the appropriate and visible Source Data for Figure 2K organized as a single source data file (zipped) per figure for main figures with the panels clearly visible in the folder structure.

>>Response: The source images for Figure 2K are now provided as 8 individual image files (one for each tile image in Figure 2K) that have appropriate color range for viewing. They have been compressed into an updated zip file.

11) As part of the EMBO Publications transparent editorial process initiative (see our policy here: https://www.embopress.org/transparent-process#Review_Process), Molecular Systems Biology will publish online a Peer Review File (PRF) to accompany accepted manuscripts. This file will be published in conjunction with your paper and will include the anonymous referee reports, your point-by-point response and all pertinent correspondence relating to the manuscript. Let us know whether you agree with the publication of the PRF and as here, if you want to remove or not any figures from it prior to publication. Please note that the Authors checklist will be published at the end of the PRF.

>>Response: We agree with the publication of the PRF as is.

13th Mar 2025

Manuscript number: MSB-2024-12595RR

Title: Multi-epitope immunocapture of huntingtin reveals striatum-selective molecular signatures

Dear Prof Cristea,

Thank you again for sending us your revised manuscript. We are now satisfied with the modifications made and I am pleased to inform you that your paper has been accepted for publication.

Yours sincerely,

Poonam Bheda, PhD
Scientific Editor
Molecular Systems Biology
